

# Climate Denial - the Antithesis of Climate Education: A Review

Gerald Kutney[1]

[1]Independent scholar, Ottawa, K1V 1Y6, Canada

*Correspondence to*: Gerald Kutney (gkutney@gmail.com)

**Abstract.** Survey after survey from across the globe suggest that climate education is floundering, despite climate education being embedded in international treaties to address the climate crisis (the UNFCCC and subsequent Paris Agreement) and the latest scientific assessment reports (IPCC) stressing the importance of climate education. The IPCC also acknowledges forces hostile to climate education, namely climate denialism sponsored by the energy-industrial complex. The latter fought the science of climate change and climate education by unleashing one of the greatest propaganda campaigns in history using

the denial machine. Climate change is studied by the physical sciences, but climate denial is the purview of the social sciences; the latter have revealed the why and how of climate denialism and the inner workings of the denial machine. A major psychological factor is individual fear among conservatives that climate change legislation represents a threat to their values and identity, and to protect their ideology they turned to climate denialism, also known as the "climate change countermovement" by sociologists. Climate-denial organizations, supported by the energy-industrial complex, are interfering

with the teaching of the science of climate change to our children. A purpose of this review is to draw attention to the growing threat of climate denialism to climate education, supported by specific examples of the influence of the energy-industrial complex in primary and secondary school classrooms.

## 1 Introduction

Climate education has a problem, a big problem, which is the focus of this review. Over a dozen years ago, Archbishop

Desmond Tutu called out for more to be done on climate education, and his comment sadly still rings true:

> Climate change is the greatest human-induced crisis facing the world today ... the tepid response of school-age and higher education to the global warming challenge, a response that focuses on technological and scientific questions while more or less ignoring the ethics of the human condition, rampant consumerism and the global marketplace ... climate change as a global justice issue (2010, xv-xvi).

These words appeared in the Forward of *Education and Climate Change* in 2010; and the editors added their own climate education call-to-arms: "At such a moment of enormous human challenge [climate change], formal, nonformal, and informal



education have a potentially crucial role to play. In both school age and adult learning communities, learners of all ages can be invited to take up the challenge of understanding and rethinking the world ... (Kagawa and Selby, 2010, 5)".

This review examines the current state of global climate education, which shows that a problem exists and considers the causes and possible solutions. The section "Results" outlines recent surveys that reflect the state of climate education. Poll after poll indicates that nations around the world are doing an insufficient job, despite climate education being an article in the international treaty on climate change (the UNFCCC) and the IPCC acknowledging the importance of climate education. The problem does not appear to be climate education directly, or climate communication or climate science. Climate denial is the external force blunting the impact of climate education. A brief overview of climate denialism, promoted and funded by the energy-industrial complex (fossil fuel and related industries), is presented, including the scientific study of climate denial and climate denialism. Specific examples of climate denial in schools are then examined in more detail. Lastly, a "Discussion" section examines the subversive role of climate denial in the classroom and how to deal with it (also, suggestions for further research are presented).

Science denial – the groundless denial of accepted science based on opinion and other unreliable sources – is as old as science itself. In the past, the practitioners of science denial were called cranks, crackpots, and junk scientists; now, a common term is science deniers. Climate denial is the most dangerous version of science denial, placing everyone at risk, and future generations even more at risk, from the global crisis. The obstructionist tools of climate denial include propaganda, misinformation, disinformation, lies, cherry-picked data, quote mining, distorted graphs, sealioning, gaslighting, fabricated evidence, false accusations of corruption (data, scientists, and organizations), discredited sources, conspiracy theories, fake news, etc. (Kutney, 2024, 26, 30, 33).

Science denial has often been driven by religious and/or political ideology; climate denial, though, was primarily driven by corporate profits. Corporate-driven science denial became more commonplace after World War II, as health and environmental sciences emerged, raising political pressure against offending industries. An early famous example of corporate-driven science denial was by the tobacco industry. With the science of climate change, the perpetrator of propaganda was the energy-industrial complex (Gelbspan, 1995, 1998; McCright and Dunlap, 2000, 2010; Oreskes and Conway, 2010, Chapter 6; Dunlap and McCright, 2015; IPCC, 2022b, 1377-1378; Kutney, 2024, 20).

Activism opposing climate denial has arisen, including by Climate Feedback (undated), Cranky Uncle (Cook, 2023a; see Figure 1), DeSmog (undated a), friends of #ClimateBrawl (Kutney, 2024, 25-27), Global Weirding (Hayhoe, undated), National Center for Science Education (see below), Skeptical Science (Cook, 2023b), by the scientists John Cook, Andrew Dessler, Katharine Hayhoe, Peter Kalmus and Michael E. Mann, and by notable news coverage in *Canada's National*



*Observer* and *The Guardian*, among others. A special mention must also be given to Adam McKay for his Academy Award-nominated political satire on climate denial, *Don't Look Up* (Netflix, 2021).

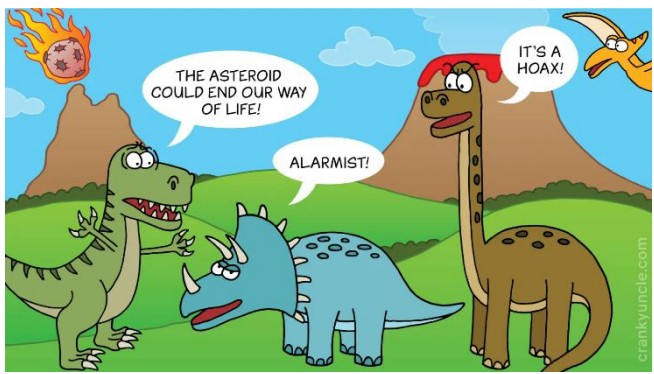
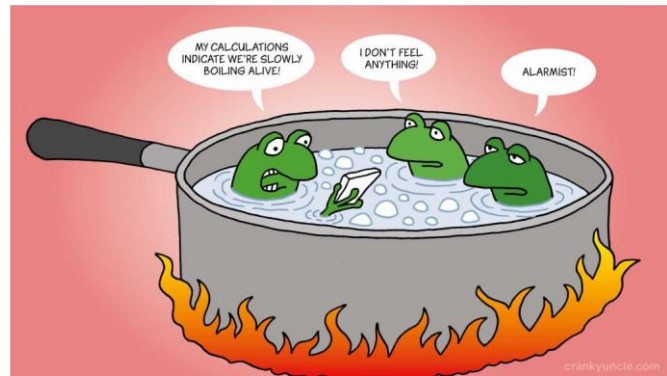

Figure 1. By permission of John Cook, crankyuncle.com.

Nevertheless, despite such praiseworthy efforts to stop it, the propaganda funded by the energy-industrial complex has
continued unabated, especially in America (Eaton and Day, 2020; Coan et al., 2021; Lewandowsky, 2021, 6; IPCC, 2022a, 1939-1940, 1982; McKie, 2022; Kutney, 2024, Chapter 5). Recently, we have seen a disturbing surge in climate denial assaults against climate education and which are described in this review.

Previous studies have described examples of the fossil fuel industrial influence in the classroom, but this review brings together recent surveys on public knowledge of the climate sciences (identifying the state of climate education), presents
specific examples of the problem with climate education, and provides a summary of the climate-denial organizations that are the leading offenders in manipulating climate education. Overall, an important aspect of the review is to create awareness of the growing threat in the classroom, so that educators and parents can protect children in their schools from climate denial. Is climate denial in your school?

## 2. Method

A chief task of climate education is the teaching of the main messages of the science of climate change to the general public and in all levels of education. This review focuses on the most vulnerable, the children in primary and secondary levels (K-12 in North America) of education. References were sought on climate denial and/or the fossil fuel industry in schools, especially those recently published in the peer-reviewed literature, with selected earlier references. Grey literature sources were added for quotes, critical commentary, and up-to-date news media information. Websites for organizations associated
with climate education and those for groups promoting climate denial in schools have also been utilized. Generally, peer-



reviewed literature was found using Google Scholar and grey literature in Google. Mainly references in English were examined.

This review sets out to answer a series of questions as follows:

- What is the current state of public knowledge of the science of climate change? To answer this question, recent surveys of public awareness on important messages from the science of climate change were examined. These polls from around the world indicated limited knowledge on the science of climate change.

- What is hindering the performance of climate education that might explain this finding? The surveys and the peer-viewed literature identified climate denial as an important source of harm to climate education. Again, recent studies were favoured but more historical information was also included to provide an understanding of climate
denialism.

- Who is promoting climate denial in climate education? The peer-reviewed and grey literature supplied direct examples of organizations promoting climate denial in schools. Most cases were found in America where climate denial is a major political force; examples were also found in Canada and Europe. Specific examples of how these organizations were promoting climate denial in climate education were found by examining their websites and
publications. Peer-reviewed and grey literature commentaries on these organizations were also examined. Greater focus was given to recent information.

In the last section ("Discussion"), conclusions, recommendations to negate the impact of climate denial on climate education, and suggestions for future research are offered. These are based mainly on the findings presented in the "Results" but also my decade-long experience challenging climate denial on Twitter (now X) and the research for my book *Climate*
*Denial in American Politics: #ClimateBrawl* (Kutney, 2024).

## 3. Results

### 3.1 The State of Climate Education

Recent surveys have revealed an alarming lack of understanding of the science of climate change by the public; such polls are also indicative of the general state of climate education itself. The polls are presented in order of the geographical scope
of the survey, beginning with one country (the United States) and ending with a survey of most countries of the world.

A survey was conducted by the Pew Research Center on how Americans viewed the understanding of climate change by climate scientists (Pasquini and Kennedy, 2023). Those who responded that climate scientists understood "very well" the causes of climate change were disturbingly low and unchanging: 28% in 2016, 28% in 2121, and 24% in 2023. There was a



major divide by political orientation: 7% of Republicans replied that climate scientists understood the causes of climate
change very well; more shocking, only 41% of Democrats had also replied very well. The authors of the study noted:

> Democrats with more education rate climate scientists' understanding higher than Democrats with less education.
> But how Republicans rate scientists' understanding of aspects of climate change does not differ by education level
> (Pasquini and Kennedy, 2023).

This poll indicated that more climate education would be beneficial for Democrats, but not for Republicans (see also
McCright and Dunlap 2011, 179; Kahn, 2015). The Pew survey shows that less than half of Americans understand the
scientific consensus and the messages of science on climate change.

A noted survey on climate change awareness among Americans by the Yale Program on Climate Change Communications
has been on-going for several years. The Yale survey breaks down replies into "Global Warming's Six Americas" (values in
brackets are the results of the poll):

#1. "Alarmed" – convinced that global warming is happening, is human-caused, and strongly support climate action
(26%).
#2. "Concerned" – convinced that global warming is happening, is human-caused, but are less motivated to take action
(27%).
#3. "Cautious" – don't know whether global warming is happening and human-caused (17%).
#4. "Disengaged" – unaware of global warming (7%).
#5. "Doubtful" – question whether global warming is happening or human-caused (11%).
#6. "Dismissive" – reject that global warming is happening and human-caused (11%).

The "alarmed" category matches best with the consensus on the science of climate change. Over the past decade, the good
news is that the share of respondents in the "alarmed" category has more than doubled (12% to 26%; and the "cautious"
category has been reduced by a comparable amount), but 26% is still not very reassuring. The bad news is that the total for
the last three categories combined has remained steady near 30% for 10 years (Leiserowitz et al., 2023a; see also 2023d; the
Yale group has also carried out a global survey using the same categories; this is presented later).

Another poll by the Pew Research Center examined Americans "most skeptical about climate change." The replies on why
they were "skeptical" included that they believed:

- natural cycles caused climate change.
- climate scientists have an agenda.





- government legislation should not restrict individual freedoms.
- climate change is a hoax (Pasquini et al., 2023).

In other words, the most skeptical just shared common climate denial talking points.

Many Americans are clearly not familiar with an important aspect of the scientific consensus that: "Human activities, principally through emissions of greenhouse gases, have unequivocally caused global warming … (IPCC, 2023, 4)". The Yale survey above had 53% of those surveyed (in the alarmed and concerned categories) agreeing that climate change was caused by human activity. The Pew Research Center also conducted a poll on public awareness of climate change in the fall of 2023 (Tyson and Kennedy, 2023; see also Tyson, Funk, and Kennedy, 2023). Only 46% replied that a "great deal" of
human activity contributed to climate change, 19% among Republicans and 71% among Democrats. Another survey on the same question found that those who replied that climate change was "mostly or entirely" caused by human activity was 49%, a drop of 11% since 2018. Democrats had dropped from 72% in 2018 to 60% in 2023, while Republicans had remained the same at 33% (Energy Policy Institute at the University of Chicago, 2023). The results of a fourth survey of American voters ("is climate change caused by humans or nature") were 59% who replied by human activity. Trust in university research
centres and scientists accounted for these low results, and there was a link to climate denialism. A recommendation to overcome these trust issues was to ensure that primary and secondary students better understand the scientific process (Alvarez, Debnath, and Ebanks, 2023).

In a survey of several European countries and the United States, the conclusion was:

We therefore advise climate change communicators, activists, and scientists to focus first and foremost on challenging the common misconception that scientists are somehow divided on the anthropogenic causes of global warming, and on closing the gap between the public and scientific consensus on climate change. In addition, specific efforts to address impact skepticism [those who believe that global warming is harmless or even beneficial] are necessary (Eichhorn, Molthof, and Nicke, 2020, 44, see also 45; see also Pasquini and Kennedy, 2023, on the scientific consensus).

In other words, these authors were warning of the negative influence of climate denial (i.e., "skepticism") on climate education.

A survey in 2022 by the Pew Research Center of 19 countries, 75% "say global climate change is a major threat to their country", ranging from 44% (Malaysia) to 86% (Greece). In addition, the survey noted political polarization in 14 of these 19 countries, where those on the right of the political spectrum were less likely to agree with the statement; the lowest
conservative responses were 22% (United States), 37% (Israel), 46% (Canada), and 47% (Australia) (Poushter, Fagan, and



Gubbala, 2022; see also Fagan and Huang, 2019.). The Pew survey on just American voters (see above, using a different question) had also found a similar divide based on political ideology.

A survey of 23 countries again showed a political divide especially in the United States. Overall, 77% agreed that: "it is essential that our government does whatever it takes to limit the effects of climate change (Marshall et al., 2023)". An interesting result was that among 82 political parties in these countries, "only 6 parties don't have majority support for pro climate policies."

In 2021, the United Nations Development Programme (UNDP) and the University of Oxford Department of Sociology released the results on the largest public-opinion survey in history on climate change with 1.2 million respondents in 50 countries: 64% agreed that "climate change was a global emergency" – results ranged from 50% (Moldova) to 81% (U.K.). Nevertheless, even among those that accepted the climate emergency, only 59% recommended doing "everything necessary and urgently" – with national results ranging from a low of 49% (Russia) to a high of 78% (Italy) (Flynn et al., 2021, 15-19) – consequently, even when people agreed "climate change was a global emergency", many were still not taking the climate crisis seriously. An initiative led by UNDP and the government of Italy, Youth4Climate, included in their manifesto a warning about climate denial from the energy-industrial complex:

> Recalling that fossil fuel companies have exercised huge power, influence and wealth, in order to intentionally spread lies, doubt and disinformation about the climate crisis for decades. This has led to widespread climate denial and "scepticism" in media and society as a whole, for the sole purpose of safeguarding the profits of their industry (Youth4Climate, 2022, 24).

A question strongly reflecting the scientific consensus of climate change was asked by Lloyd's Register Foundation in partnership with Gallup. In their poll of 121 countries, the question was asked (emphasis added): "Do you think that climate change is a very serious threat, a somewhat serious threat, or not a threat at all to the people in this country in the **next 20 years**?" Only an average of 67% chose a very serious threat or somewhat serious threat; results varied from 31% (Saudi Arabia) to 94% (Chile). Among the factors effecting public perception of the climate crisis was: "Disinformation campaigns and vested interests seeking to downplay the severity of the problem further muddy the waters, sowing doubt and confusion (Lloyd's Register Foundation, 2023, 2; see also 2022, 46; see also Kennedy, 2023; Poushter, Fagan, and Gubbala, 2022);" in other words, climate denial was hindering knowledge building on the science of climate change and the scientific consensus.

The broadest survey of almost 200 countries (no data were available from China, Russia, and Iran) has been undertaken by the Yale Program on Climate Change Communications (see above) and Meta in two reports (Verner et al., 2023; Leiserowitz et al., 2023c). Using the same categories as for their American surveys, a selection of countries in the "alarmed" and "dismissive" categories is presented in Table 1 to highlight the range of national differences.



| Country | "Alarmed" | "Dismissive" | Country | "Alarmed" | "Dismissive" |
|---|---|---|---|---|---|
| Chile | 65% | 0% | United Arab Emirates | 39% | 4% |
| Mexico | 64% | 0% | Japan | 36% | 2% |
| Sri Lanka | 61% | 0% | Canada | 35% | 5% |
| Brazil | 59% | 1% | Australia | 35% | 6% |
| Kenya | 55% | 5% | United States | 34% | 11% |
| India | 55% | 2% | Germany | 34% | 3% |
| Turkey | 51% | 3% | United Kingdom | 31% | 4% |
| South Africa | 49% | 2% | Saudi Arabia | 29% | 4% |
| Spain | 47% | 1% | Sweden | 27% | 5% |
| Italy | 45% | 1% | Norway | 13% | 8% |

Table 1. Percentage of survey respondents from selective countries who are convinced that global warming is happening, is human-caused and an urgent threat ("alarmed" category) and those who reject that global warming is happening and human-caused ("dismissive" category) (Verner et al., 2023).

The "alarmed" category best represents the scientific consensus, but only 29 out of almost 200 countries polled at 50% or greater. Apparently, few people anywhere are getting the full message of the science of climate change or the scientific consensus. Any category below "alarmed" indicates some degree of climate denial, with the most extreme climate deniers found in the category of "dismissive" (only the United States was greater than 10% in this category).

In the second report by the Yale Program on Climate Change Communications and Meta representing 187 countries, the
majority in only 16 countries agreed that climate change is "mostly caused by human activities (Leiserowitz et al., 2023c, 8, 30-35)". A majority in most countries did agree that climate change should be high priority for their government (Leiserowitz et al., 2023c, 13, 60-65).

Decades of scientific evidence have created a scientific consensus that climate change is human-caused and is now an urgent threat, which has been affirmed by the high-profile, global assessments of the peer-reviewed literature by the IPCC (2023, 4, 
24). Yet, poll after poll have found that much of the global public are ignorant of the irrefutable messages of the science of climate change and the scientific consensus. This "Consensus Gap (Skeptical Science, undated)" between the public and science haunts policy development on the climate crisis. Until the public accepts the basic tenets of the science of climate change, legislation to address the climate crisis is unlikely, if not impossible. This Consensus Gap has been created by the energy-industrial complex through climate denial.





## 3.2 The UNFCCC and IPCC on Climate Education


Better results from the global surveys would have been expected given that climate education is a treaty obligation. Under the United Nations Framework Convention on Climate Change [(UNFCCC, 1992, Art. 6); see also the Kyoto Protocol (1998, Art. 10e) and the Paris Agreement (2015, Art. 12)], 198 countries are obligated to develop climate education programs. Recently, the UNFCCC has reiterated its importance:

> Climate change education is one central foundation to achieve the goals of the Paris Climate Change Agreement. It can provide everyone – children, youth and adults ... Education about climate change, above all for young people, is presently sorely lacking on a global scale (2023; see also 2022).

and a declaration at COP28 reminded delegates of their responsibilities:

> Recalling Article 6 of the United Nations Framework Convention on Climate Change and Article 12 of the Paris
> Agreement, we call upon countries to enhance climate change education to support transitions to low-carbon and climate-resilient economies and societies (UNESCO, 2023b).

The UNFCCC concern about climate education is supported by a poll of the United Nations Educational, Scientific and Cultural Organization (UNESCO) in a survey on 100 countries integrating climate education into their curriculum. UNESCO introduced its study by stating: "This document begins with the assumption that education is essential to prepare societies to
address the climate crisis ... there is a need to understand the depth of inclusion of climate change education within national curriculum frameworks (2021, 4)". The overall finding of the survey was that 93% of countries had no or a very minimal level of content on climate change in their national curriculum (UNESCO, 2021, 12). Another UNESCO study found that:

> The quality of the current climate change education is in question. Seventy per cent of the youth surveyed say that they cannot explain climate change, or can only explain its broad principles, or do not know anything about it,
> putting into question the quality of climate change education in our schools today (UNESCO, 2022a, 3).

Another article of the UNFCCC states that the Intergovernmental Panel on Climate Change (IPCC) will provide "objective scientific and technical advice (1992, Art. 21.2)". The IPCC presents the consensus view of the state-of-the-art of the science of climate change in their assessment reports (Oreskes, 2004). A special aspect of the IPCC assessment process on the
scientific consensus is the approval of the reports by global governments (IPCC, 2021). The science of climate change is the most scrutinized science in history because of the IPCC assessments (and others). Considering the scientific consensus and the robust nature of the science, it is therefore all the more puzzling that climate education has not been more successful.



Psychologist Dan Kahan bluntly described the frustrating problem: "The failure of widely accessible, compelling science to quiet persistent cultural controversy over the basic facts of climate change is the most spectacular science communication
failure of our day (2015, 2)".

The latest reports of the IPCC (AR6) have extensively commented on education, mainly in *Climate Change 2022: Impacts, Adaptation and Vulnerability* (2022a) and *Climate Change 2022 – Mitigation of Climate Change* (2022b). There are two statements on climate education that survived the gruelling Summary for Policymakers (SPM) approval process (i.e., only the most salient points appear in the SPM); first, in *Impacts, Adaptation and Vulnerabili*ty:

245         Enhancing knowledge on risks, impacts, and their consequences, and available adaptation options promotes societal
            and policy responses … sources can deepen climate knowledge and sharing, including capacity building at all
            scales, educational and information programmes (IPCC, 2022a, 28),

and the second, in the important *Synthesis Report*:

            Increasing education including capacity building, climate literacy, and information provided through climate
250         services and community approaches can facilitate heightened risk perception and accelerate behavioural changes
            and planning (IPCC, 2023, 30; see also 107).

The IPCC has also acknowledged a hostile countermovement against climate education and the science of climate change through a variety of related terms mentioned in the AR6:

- climate denial (2022b, 185, 469, 526, 1737)
- scepticism (2022b, 469, 524, 555, 1374, 1377, 1737)
- climate change countermovement (2022b, 58, 127, 557, 1358, 1377)
- contrarian (2022a, 1940)
- misinformation/disinformation (2022a, 954, 1931, 1939, 1940, 1982, 2712; 2022b, 58, 1377, 1411).

These terms are all associated with climate denial.

In the Technical Summary of *Climate Change 2022 – Mitigation of Climate Change*, the pernicious effect of climate denial on climate education is clearly presented: "Accurate transference of the climate science has been undermined significantly by climate change counter-movements … through misinformation (IPCC, 2022b, 58; see also 1377; 2022a, 1931, 1939, 1940)". A similar warning appeared in the *Synthesis Report*: "... organised counter movements have impeded climate action, exacerbating helplessness and disinformation and fuelling polarisation, with negative implications for climate action (IPCC,



2023, 52)". In summary, the AR6 reports of the IPCC acknowledge how climate education has been adversely impacted by the propaganda of climate denial.

The IPCC has also reported on the involvement of the oil industry (and other members of the energy-industrial complex) in these climate denial campaigns:

Vested interests have generated rhetoric and misinformation that undermines climate science and disregards risk and urgency ... Resultant public misperception of climate risks and polarised public support for climate actions is delaying urgent adaptation planning and implementation (2022a, 1931)",

and "the oil industry has underpinned emergence of climate scepticism (2022b, 1374)".

Yet, awkwardly, the COP28 of the UNFCCC was hosted by the United Arab Emirates and the president-designate of the convention was the chief executive of the Abu Dhabi National Oil Company who caused much outrage with his comments
"verging on climate denial (Carrington, 2023)", and an ex-oil executive, Mukhtar Babaye, has been appointed president for COP29 in Azerbaijan (Gayle, 2024). We cannot then be surprised to find that the energy-industrial complex is embedded in climate education, attempting to undermine the teaching of the science of climate change in schools.

### 3.3 Climate Denial

The first major exposé on climate denial appeared in the mid-1990s by the journalist Ross Gelbspan. He described the
disinformation campaigns by the energy-industrial complex (fossil fuel and related industries) to cast doubt on the science of climate change in the news media and government hearings (Gelbspan, 1995, 1997). A decade later, another investigative report appeared on Canadian TV, on the CBC public affairs program the *Fifth Estate*. The "Denial Machine" revealed how the energy-industrial complex followed the tobacco strategy and financed the "denial machine", including contrarian scientists and PR firms to dupe the public into thinking that there was still a debate about the science of climate change (the
original broadcast does not exist on-line, and requests to CBC went unanswered; see Government Accountability Project, 2006). A year later, the "denial machine" was back in the news in an article in *Newsweek*, titled "Global Warming Deniers Well Funded". The article described how the "denial machine" was framing public opinion and killing climate bills in Congress through propaganda by contrarian scientists, right-wing think tanks and industry creating a "paralyzing fog of doubt around climate change (Newsweek Staff, 2007)".

Scholarly studies have also provided overwhelming evidence of climate denial funded by the energy-industrial complex against climate science and climate education.





### 3.3.1 The Science of Climate Denial

A masterful study on anti-environmental propaganda by corporations appeared in 2010 – *The Merchants of Doubt*. The first chapter of the book by historians Naomi Oreskes and Erik Conway was called "Doubt is Our Product (2010, Chapter 1)".

Here, the authors presented the disinformation campaigns of the tobacco industry, and then gave similar accounts in other industries. Many of the major science denial actors for tobacco were hired by the energy-industrial complex, including the same right-wing think tanks and contrarian experts (Oreskes and Conway, 2010, Chapter 6). There was no connection with the science of smoking causing cancer and the burning of fossil fuels causing climate change, but knowledge of the science was not what mattered; the main job requirements were being good at propaganda and denying science.

The science of climate change is essentially a physical science, but the study of climate denial is a subject for the social sciences. Individual denial of the science of climate change is more a topic for psychologists who deal with "climate denial", whereas collective denial of the science of climate change is explored more by sociologists who study the "climate change countermovement" or "climate denialism".

Denial, itself, is a psychological defence mechanism for dealing with stress and fear. The concept was introduced by

Sigmund Freud and was studied further by his daughter Anna Freud in *The Ego and the Mechanisms of Defense* (1937, Chapters 6-7). By the turn of the millennium, sociologists had found climate denial to be a social "countermovement" (McCright and Dunlap, 2000). An early report specifically on the psychology of climate denial appeared in 2001. The study examined "socio-psychological denial mechanisms". The authors found that: "The most powerful zone for denial was the perceived unwillingness to abandon what appeared as personal comfort and lifestyle-selected consumption and behaviour in

the name of climate change mitigation (Stoll-Kleemann, O'Riordan, and Jaeger, 2001, 113)". In a second study two decades later, a dichotomy had arisen. The majority accepted the serious nature of climate change and supported "mitigation in the abstract", but were still not doing much individually, waiting for others to act first (Stoll-Kleemann and O'Riordan, 2020, 12; see also Wullenkord and Reese, 2021).

Several cognitive biases in climate denial and ways of overcoming them have been reviewed (e.g., Zhao and Luo, 2021).

Psychologists have found that prebunking is more effective than debunking in the case of the climate propaganda (Lewandowsky et al., 2020; Lewandowsky, 2020, 11-12); in short, climate denial talking points are best discredited pre-emptively. In November 2022, a collaborative publication between the journals *Nature Human Behaviour* and *Nature Climate Change* was called "Climate change and human behaviour" (Antusch and Yan, 2022). In this series, Hornsey and Lewandowky (2022) examined the psychological origins of climate denial and how to challenge it (see also Wong-Parodi

and Feygina, 2020; Ekberg et al., 2023). Also, in this special *Nature* issue, an article by Jenny and Betsch explained that: "improving individual knowledge through better communication alone is insufficient (2022)" and that industries must be targeted to deal with the climate crisis.





Climate denial arises from fear of science messages about climate change, especially among conservatives. Political ideology plays a lesser role in climate legislation outside the United States (Hornsey, Harris, and Fielding, 2018; for the degree of climate denial outside of America, see also Dunlap and McCright, 2015, 318-320; McCright, 2016, 79-82; Eichhorn, Moltof, and Nicke, 2020, 16, 45; Hornsey and Lewandowsky, 2022; Nartova-Bochaver et al., 2022; McKie, 2022).

The fear of conservatives, though, is not the existential threat from climate change, but from being a perceived threat to the worldview of conservatives and their traditional American values, especially individual freedom, and free enterprise (McCright and Dunlap, 2000, 504-505). McCright and Dunlap studied over 200 climate denial reports from 30 right-wing think tanks between 1990 and 1997 (notably, most of the reports appeared in 1997, the year of Kyoto). Two arguments were present in more than half of the think tank reports: the science of climate change is highly uncertain (63%) and climate change policies would harm the national economy (58%) (McCright and Dunlap, 2000, 510-518). In 2010, McCright and Dunlap reviewed again the opposition of conservatives to environmental issues as a countermovement called "a force of anti-reflexivity" to defend the "industrialist social order (2010, 104)". Later McCright wrote that: "The leading theoretical explanation for the mobilization of organized climate change denial is the Anti-Reflexivity Thesis, which characterizes the climate change denial countermovement as a collective force defending the industrial capitalist system (2016, 77; see also Brulle, 2014; Dunlap and McCright, 2015)".

Climate education is important to resolving the climate crisis; as psychologist Stephan Lewandowsky has reported: "people's knowledge about climate change matters (2020, 10)". However, barriers stand in the way; namely climate denial that is serving vested interests, such as hard-line conservatives and the energy-industrial complex (Oreskes and Conway, 2010, Chapter 6; McCright and Dunlap, 2011; Dunlap and McCright, 2015; Hornsey, Harris, and Fielding, 2018; Lewandowsky, 2021; Hornsey and Lewandowsky, 2022; Kutney, 2024). Stephan Lewandowksy has warned: "These political implications have created an environment of rhetorical adversity in which disinformation abounds, thus compounding the challenges for climate communicators (2020, 1)", and "The terrain for climate communications is treacherous (2020, 8)" because of the adversarial environment created by climate denial. Stoetzer and Zimmermann reached similar conclusions:

> If protecting one's [political] group identity outweighs other motives, then from a policy perspective, reducing the existing misperceptions will be a difficult task. The key challenge would be to change group identities or weaken them altogether, which seems uncharted territory for policymakers (2024).

These political ideological barriers to climate education are formidable. McCright and Dunlap (2011, 179) and Kahn (2015) found that climate education had little impact on conservatives in America (see also surveys above), again related to climate

denial. In summary, public opinion surveys, investigative reporting by the news media, and social science studies support the premise that climate education is under siege by a powerful foe – climate denial, the antithesis of climate education.

## 3.4. Climate Denial in the Classroom

An early study about science denial in the classroom was a portent to climate educators:

> In this feature, I began by considering organized and intentional denialism, about which every honest scientist and educator must be concerned … Teachers and students who recognize the role of science in our society should be able to recognize a denialist tactic when they see it (Liu 2012, 134).

Over a decade later, the situation on climate denial in the classroom has become more dire. Is climate denial in your
classroom?

Looking directly at primary and secondary schools, a pattern unfolds of a softer climate denial infiltrating the K-12 (kindergarten to grade 12) classrooms, especially in STEM (science, technology, engineering and mathematics) education. Such "petro-pedagogy" is less overt than traditional conservative climate denial, but it still ignores the harmful role of its content providers – the energy-industrial complex – in the creation of climate change while excessively promoting the
virtues of fossil fuels (Eaton and Day, 2020; Tannock, 2020). A general relationship has unfolded:

Charity-based education programs + energy-industrial complex sponsorship → petro-pedagogy

Beware of the energy-industrial complex bearing gifts. Petro-pedagogy is a Trojan Horse with climate denial stealthily hidden within and brought willingly into the classroom, whereby children and teachers are converted into fossil fuel enthusiasts. Petro-pedagogy teaches that oil is a benefactor to humanity, and modern civilization cannot exist without fossil
fuels, but says little, if anything at all, about the connection of fossil fuels to the climate crisis. This newer expression of climate denial is one also used by "oil apologists" who laud fossil fuels by exaggerating how indispensable their contribution is to society yet are silent on their negative impact on the climate – this is climate denial by omission (Kutney, 2022).

Below is a description of conservative organizations and petro-pedagogy organizations promoting climate-denial in the classroom (see also figure 2 for editorial cartoons reflecting both groups). The energy-industrial complex is the sponsor
behind these organizations.





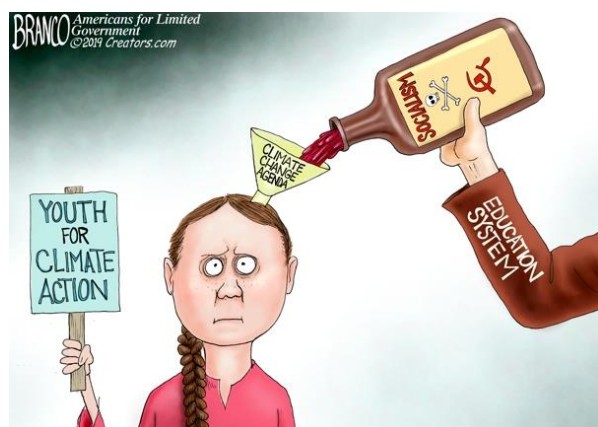
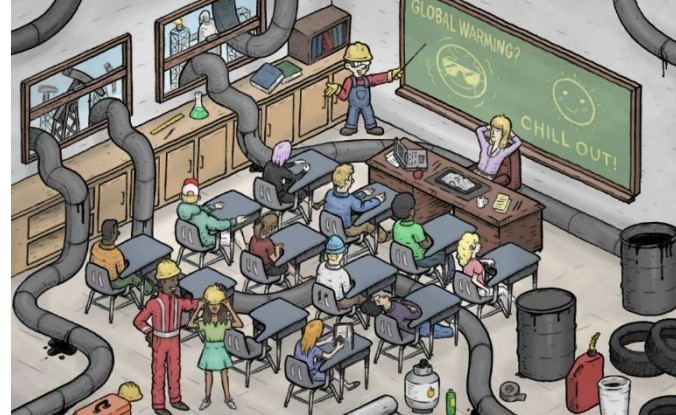

Figure 2. L - By permission of A. F. Branco and Creators Syndicate, Inc.; R - By permission of Eben McCue (ebenmccue.com/).

### 3.4.1 Global

### 3.4.1.1 Energy4me

An example of such a petro-pedagogy program that offers programs around the world is Energy4me, offered by the Society of Petroleum Engineers. Their homepage states: "Energy4me promotes fact-based education to help demystify the industry. It is designed to promote an energy conscious and educated society, and create interest in science, technology, engineering and math (STEM) careers (Energy4me, 2024a)". On their Sustainability page, they acknowledge the movement away from

fossil fuels: "Though the world is reducing its footprint and moving towards more sustainable and cleaner energy, it will be a slower transition than most expect (Energy4me, 2024b)". Their Energy Transition page recommends switching from coal to natural gas, adding: "But we can't survive on renewables alone. Forward-thinking oil and gas companies are already considering ways to optimize their portfolios to avoid stranded assets and are pursuing opportunities in gas and renewable energy (Energy4me, 2024c)". On their Environmental Protection page, Energy4me briefly discusses global climate change

including that the production of fossil fuels produces emissions that cause the greenhouse effect but ignores that the burning of fossil fuels as a major cause (2024d). There is also a video on this webpage from Switch Energy Alliance (see the next entry). Energy4me does cover climate change better than most petro-pedagogy sites, but the direct influence of fossil fuels on the climate crisis is still lacking.



### 3.4.1.2 Switch Energy Alliance

Switch Energy Alliance is known for its films and videos promoting fossil fuels globally. The chairman and founder of the group is Scott Tinker, a professor of geology at the University of Texas. In 2019, Dr. Tinker wrote an article that began with:

> Republicans often get branded as "climate-change deniers." Most don't deny natural climate change, but many do question the extent of human responsibility for present-day warming and of threatened catastrophic effects. By contrast, Democrats often get labeled as "climate solution deniers." Democrats propose to get off fossil fuels and
> switch to renewables as fast as possible — although, unfortunately, neither of these things will have much impact on climate change (2019a).

Then in *Scientific American*, Dr. Tinker criticized carbon pricing and renewables, and stated: "Unfortunately, those who are the most passionate about addressing climate change seem to not like the answers from the energy experts (2019b)". He, then, suggested that the answers were natural gas, nuclear energy, and carbon capture. In 2023, Dr. Tinker then advocated
fossil fuels to solve energy poverty and criticized those who focused on a "climate catastrophism narrative (2023)".

Switch Energy Alliance is: "dedicated to inspiring an energy-educated future that is objective, nonpartisan, and sensible (2024a)". One of their programs, Switch Classroom, "provides innovative tools and expert-driven content to enable students to think critically about energy (Switch Energy Alliance, 2024b)". No information was found on the importance of reducing the burning of fossil fuels to resolve the climate crisis, but criticisms were often raised about renewable energies.

### 3.4.2 Europe

### 3.4.2.1 BP Educational Service/Energising Futures

A notable organization in petro-pedagogy is BP, which: "has successfully embedded itself at the heart of elite U.K. science and education policy and practice networks ... (Tannock, 2020)". BP Educational Service (BPES) has had a significant influence on the U.K. school system:

• 84% of U.K. schools are registered to BPES.
   • >100,000 teachers are registered to BPES.
   • >1.7 million students have used BPES (BP, 2023).

BPES has recently reinvented itself as "Energising Futures" (BPEF) which was launched on 20 February 2023.

BP works with the Association for Science Education (ASE) on its school programs. The ASE site had a webpage on key
"collections" from BPES, including "Climate.Speaks" which: "introduces arguments positioned through different



stakeholders such as government, activists, energy and transport companies, and agriculture (Association for Science Education, 2020)". However, Climate.Speaks no longer exists in the BPES or BPEF websites. Not long after I contacted the ASE about Climate.Speaks no longer being on the BP websites, their webpage was taken down. There is very little information on climate change in the new BPEF anywhere. The information that is available is a two-page pamphlet for

students aged 14 to 16 called "Earth's atmosphere knowledge organiser" which mentions fossil fuels contributing to climate change (BP, 2024a), and a renewable energy pamphlet mentioning the Paris Agreement and that fossil fuels "contribute to climate change" (BP, 2024b). In short, climate change and its connection to the burning of fossil fuels are given a perfunctory recognition. Tannock found that the petro-pedagogy of BPES: "poses a significant threat to our collective efforts to tackle the global climate crisis (2020)". There is no indication that BPEF will be any better.

**3.4.2.2 Scientix**

The interaction of oil industry representatives with secondary teachers and students is supported by the European Union (Andrée and Hansson, 2023, 2, 5). Andrée and Hansson looked at Scientix (EU-funded program for advancing STEM teaching) promoting careers in the petroleum industry for secondary science students. Their webinars (Scientix, undated) on career paths produced in collaboration of the European Petrochemical Association (EPCA) were examples of petro-

pedagogy by promoting the petrochemical industry as responsible corporate citizens that were essential to modern society and saving the world from global environmental crises (yet, distancing themselves from the role of fossil fuels in the climate crisis). The European petrochemical industry has a problem with recruitment: "an awareness of the industry's direct impact on climate change is leading young people to believe there is no long-term future in the industry (European Petrochemical Association, 2023)". Researchers concluded that:

the petrochemical industry representatives communicated petro-pedagogy interests, beliefs and narratives directly to students participating in classrooms across Europe … it might be difficult for teachers as well as policy makers to see through the 'smokescreens' of the webinars (Andrée and Hansson, 2023, 13).

**3.4.2.3 Shell's *It's All About Energy* and NXplorers**

Shell created the NXplorers program on STEM education, which is offered around the world, including The Netherlands (undated). In a video about the program, the Project & Technology Director of Shell described: "… sustaining a liveable climate. To address this vastly complex question, we need carefully, well-thought-thorough solutions that focus on the long term (Shell, 2018)". Shell is promoting climate change but only as a long-term issue.



Petro-pedagogy in The Netherlands has been criticized by the group *Fossielvruij Onderwijs* ("Fossil Fuel Education") and its
sister organization *Reclame Fossielvrij* (Advertising Fossil Free). Shell, for example, had produced in 2012 a textbook *It's
All About Energy* for Dutch secondary schools (and later translated into ten other languages), where carbon capture and
storage and natural gas were promoted as solutions to the climate crisis (*Reclame Fossielvrij*, 2021). The teaching materials
had been distributed by the Dutch organizations Jet-Net (part of the EU STEM coalition) and its partner InGenious (to
coordinate STEM education in Europe, launched by European Schoolnet and European Round Table of Industrialists).
InGenious was concluded in 2014 but its website is still active, including a page on *It's All About Energy*:

> The final section looks at technological solutions for dealing with $CO^2$, taking in such diverse concepts as phase
> states of compounds, questions of safety and societal impact. As with the other sections, the practice introduces the
> key concepts behind the current thinking on the subject, explains technical ideas in straight-forward language, and
> suggests different scenarios and solutions.

> Best of all, it encourages students to think creatively about how they might tackle the challenges of climate change
> and emissions – rather than simply spoon-feeding them ideas (undated).

Jet-Net, too, still has a webpage dedicated to *It's All About Energy* but links on the site were broken (undated). Shell
provides a map to links to their educational programs around the world. For The Netherlands, there is mention of Shell's
support for Jet-Net, but the link is broken (Shell, undated). There is another broken link on the Shell site to the Generation
Discover Festival in The Netherlands (undated). The Generation Discover Festival has been accused to be Shell
greenwashing (*Fossielvruij Onderwijs*, undated a).

*Fossielvruij Onderwijs* criticized the educational programs of the Dutch energy-industrial complex, such as those of Shell:
"Whatever they teach children, it is fundamentally wrong for them to be in front of the classroom. The fossil industry puts its
own business model above the interests of these children to grow up safely and healthily. They present children with a
worldview that is already outdated (undated b)". The energy-industrial complex of The Netherlands was promoting a slow
transition, and that fossil fuels would be needed for a long time. *Fossielvruij Onderwijs* placed Shell second in a ranking of
Dutch climate deniers because of their "misinformation campaign aimed at children (undated a)".

### 3.4.3 North America

### 3.4.3.1 Canada

Concern has been raised about climate education in Saskatchewan, where the Canadian energy-industrial complex had
funded energy and climate education programs for K-12 education (Eaton and Day, 2020, 462). All of these "petro-
pedagogy" initiatives promoted the notion that education on climate change must prioritize the benefits of the fossil fuel



industry. The training courses taken by teachers established more favourable views of the energy-industrial complex which reflected what the teachers taught in the classroom in some cases (Eaton and Day, 2020, 465-469). The same researchers concluded that there was an "urgent need to dismantle the corporate power of the fossil fuel industries and their petro-pedagogy (2020, 470)". Examples of petro-pedagogy in Canada are discussed below.

### 3.4.3.1.1 FortisBC Energy Champions

FortisBC reports that: "Our school programs are delivered together with BC teachers, FortisBC employees and external organizations such as the BC Lions [Canadian Football League team in Vancouver]. Our classroom-ready materials help teach students, from kindergarten to high school, about how to be safe around energy and ways to conserve it (2024)". Their programs include Energy is Awesome for grades two to five (the company has suspended this program) and Energy Champions for K-7 in partnership with the popular CFL BC Lions promoting the program. Energy Champions is on the importance of environmental responsibility with a focus on energy conservation. The FortisBC website, though, provides little detail about their educational programs.

A review of climate education by FortisBC criticizes their programs as a "sales pitch" by the company (Cruickshank, 2022; see also Gamage 2022). Emily Eaton had been quoted in the article: "But what many people have called the new climate denialism is this idea that we're actually denying the scale and speed, or pace, of the changes that are needed to rescue a habitable planet. And so that's the kind of denial that I see in these types of resources" (Cruickshank, 2022). Similar views to delay climate action were cited in the article by the climate communication and climate denial expert, John Cook.

### 3.4.3.1.2 Inside Education

An Alberta-based organization Inside Education has a mission to: "support teachers and inspire students to better understand the science, technology, and issues related to our environment and natural resources (2024)". Part of their program "Stewardship, Energy, Climate & You" includes a Teacher's Guide which provides a reasonable enough snapshot of climate change on the surface, including the increase in extreme weather events associated with it. However, the Guide promotes personal action, and not collective action or policies to restrict the production of fossil fuels (Inside Education, 2018; see also Eaton and Day, 2020, 465; Hodgkins, 2014). Inside Education has been promoted by the Canadian Energy Centre, which is sponsored by the Alberta Government (CEC Staff, 2023).

### 3.4.3.1.3 Pathways Alliance

Pathways Alliance, an association of major oil sands companies in Canada (Pathways Alliance, 2024), does not have a specific education program on their website. However, in early-2023, the Central Alberta Teachers' Convention Association hosted a presentation on Pathways Alliance at their annual meeting (Meyer, 2023).



### 3.4.3.1.4 Safety in Schools Foundation

On December 7, 2023, the Safety in Schools Foundation launched their Energy Career Literacy program (2023). They are developing programs for grades K to 6 and 7 to 9. In a video address, Brian Jean, the Minister of Energy and Minerals for Alberta, congratulated the Safety in Schools Foundation for their new initiative (Safety in Schools Foundation, 2024). There were few details available about the new initiative.

### 3.4.3.1.5 SEEDS

Another fossil fuel industry sponsored organization, SEEDS Connections, provided: "educational programs related to leadership, environment, energy and diversity, for Kindergarten to Grade 12 students across Canada ... and distributes this program to over 2,000 schools across Canada (2019a)". There is a webpage "Teaching Activities for Climate Change" which only promotes energy saving and completely ignores the problem with burning fossil fuels (SEEDS Connections, 2019b). Their website has not been updated for the last few years. SEEDS has been accused of promoting propaganda in schools (Eaton and Day, 2020, 463, 466).

### 3.4.3.1.6 Ten Peaks Innovation Alliance

Ten Peaks describes itself as a: "not-for-profit with a mission to engage, inspire, and educate Alberta's youth about energy, the environment, and our climate and how they can play an essential role in the future of our province (undated)". Not much information on climate change was found in their website, but there was much related to fossil fuels. In their first blog introducing the group, the following statement was made:

> Working in the [oil and gas] industry for more than 25 years, [founder and Executive Director of Ten Peaks Dagma Knuston] knew how far the industry had come and its commitment to addressing climate change. Every day she witnessed Alberta's oil and gas professionals putting energy, the environment and climate at the forefront of the industry …This information is not known to many Albertans, let alone Alberta's next generation of leaders. (Ten Peaks, 2020).

The location of many of the above organizations demonstrates that petro-pedagogy in Canada is Alberta-centric, which comes as no surprise since this province is the center of the oil and gas sector. More information on climate education in Canada can be found in "Climate Change Education within Canada's Regional Curricula: A Systematic Review of Gaps and Opportunities (Field et al., 2023; see also 2019)", and Canada's entry in Profiles Enhancing Education Reviews, PEER (UNESCO, 2022b).





### 3.4.3.2 United States

A survey found that 75% of registered voters in the U.S. supported the idea: "schools should teach children about the causes and consequences, and potential solutions to global warming;" the results varied by political affiliation:

- Liberal Democrats – 98%
- Moderate Democrats – 91%
- Moderate Republicans – 77%
- Conservative Republicans – 40% (Leiserowitz et al., 2023b).

Political initiatives have been attempted at the federal level on climate education, namely the Climate Change Education Act which had been first introduced by Senator Barack Obama and Representative Michael Honda in 2007 (under the name of the Global Warming Education Act), which was re-introduced by Senator Edward Markey (2021) and Representative Debbie Dingell (2021); however, no further progress in Congress has taken place.

Education in America is decentralized and generally controlled by the state. Climate Changemakers have initiated an email campaign to have climate change included in the state education standards and to teach educators about the climate crisis (undated).

Climate education is included in the important K-12 content of Next Generation Science Standards (NGSS; a collaborative, state-led process of 26 states), which recommends the teaching of climate change beginning in middle school, ages 11-13:

> Human activities, such as the release of greenhouse gases from burning fossil fuels, are major factors in the current rise in Earth's mean surface temperature (global warming). Reducing the level of climate change and reducing human vulnerability to whatever climate changes do occur depend on the understanding of climate science, engineering capabilities, and other kinds of knowledge, such as understanding of human behavior and on applying that knowledge wisely in decisions and activities (2017, 59).

The NGSS has been adopted by 20 states, and 24 others are using them as guides (National Center for Science Education, 2020, 2) for their science curriculum, leaving only six states not using the NGSS to some extent at least.

The National Center for Science Education (NCSE) and the Texas Freedom Network Education Fund prepared a state-by-state report card on public school science standards on climate change. The grade of the 20 states which followed the NGSS was a "B+", and 10 states received poor grades (those in bold were not guided by the NGSS): "D" grade – **Florida**, Indiana, **Ohio**, and West Viriginia, and "F" grade – Alabama, Georgia, **Pennsylvania**, South Carolina, **Texas**, and **Virginia** (National Center for Science Education, 2020).



Previously, in 2015, an extensive survey was conducted on climate education in American schools by the NCSE. Among science teachers in middle and high schools, 75% discussed global warming for at least one class, but almost a third were

teaching that according to many scientists, recent global warming was "likely due to natural causes (Plutzer et al., 2016a, 15-16; see also 2016 b)". The NCSE warned: "Teachers, administrators, and community members must remain vigilant against efforts to introduce denial into classrooms ... Owing to organized efforts by climate change deniers, there is a wealth of well-presented misinformation available online and in some cases mailed directly to teachers (Plutzer et al., 2016a, 33)". The NCSE also has a webpage on "The Pillars of Climate Change Denial", which provides information for challenging climate

denial because it is: "critical to defend the teaching of climate science (National Center for Science Education, 2016)". In 2023, the NCSE harshly criticized the deceitful tactics of climate denial organizations: "Cartoons and jokes and lies: that's the recipe for climate change denial aimed at kids, sometimes kids as young as six years old, judging from recent campaigns from conservative outfits ... Whatever its source, it's dismaying that such propaganda is aimed at so young an audience (Reid and Branch, 2023)".

A study of climate education in the United States found that all states had policies mentioning climate change, but 33 states had very low focus, and 14 states had low focus on climate change content (MECCE and NAAEE, 2022, 9-11, 24, 40). The report again highlighted the issue of climate denial: "For decades, political and social will to act on climate change was quickly swept away in a current of denial, avoidance, and political posturing (MECCE and NAAEE, 2022, 3; see also 7, 28, 44)".

In "Miseducation, How Climate Change is Taught in America", investigative reporter Katie Worth identified case after case of interference in American schools by the energy-industrial complex. Worth writes about the: "intentional miseducation of our children (2021)".

A video by Climate Town presents an excellent overview of "The Brainwashing of America's Children". Early in the video, the narrator Riley Williams comments:

The oil and gas industry has spent millions of dollars trying to influence American school children … there is a massive paper trail of oil-funded lesson plans and workbooks. They produce propaganda videos and pro fossil fuel cartoons … and a whole gaggle of shady tactics to push their agenda on kids … these efforts to influence what children hear in public schools seem to be working (Climate Town, 2023).

A history of the oil industry propaganda and climate denial in schools is presented in the video by Climate Town. Below are

organizations of petro-pedagogy and conservative climate denial infiltrating American schools (for more on American climate education see Bhattacharya, Steward, and Forbes, 2021; UNESCO, 2023a).



### 3.4.3.2.1 CO2 Coalition

A well-known conservative climate denial group is the CO2 Coalition (DeSmog, undated b). Their latest "educational comic book" is described as:

Once Upon a Time: A true story about the miracle molecule--carbon dioxide provides scientific information in a manner that is simple enough that even a young child can enjoy and understand. The story is told through the adventures of three young girls who live on a tree farm in Oregon. They learn from their kindly neighbor, a scientist, that carbon dioxide (CO2) is the miracle molecule that is necessary for life on earth to exist and that increasing CO2 is helping plants to grow faster and bigger (CO2 Coalition, undated a).

The Lesson Plan for the comic book includes the following:

     For years, innocent children have been terrified by threats of harm caused by humanity's use of fossil fuels. The supposed evil villain of this fairy tale, atmospheric carbon dioxide, is in fact beneficial to life on Earth. These illustrated stories give an entertaining and scientifically accurate explanation of why carbon dioxide is the 'gas of life' and why we and other living things are lucky to have more of it (CO2 Coalition, undated b, 2).

On the webpage for this comic book, a comment is made on how children have been taught incorrectly about the molecule, as CO2 is not the "demon molecule" but the "miracle molecule (CO2 Coalition, undated a)". A request to CO2 Coalition for permission to post the cover cartoon of "Once Upon a Time" in this review on climate education received no response.

### 3.4.3.2.2 EverBright Media

EverBright Media, founded by former Arkansas governor Mike Huchabee, claims that: "more than 700,000 families are
enjoying [their] products (undated a)". Their pamphlets, decorated with cartoon covers, include the "Kids Guide" to "Free Markets", "Fighting Socialism" and "The Truth about Climate Change (undated b)". The Arkansas Department of Education has purchased "The Kids Guide to Coronavirus" from EverBright, but not yet "The Truth about Climate Change". Regarding the latter, issued in 2023, one commentary concluded: "they deliberately undermine children's scientific education ... They're not just trying to create climate skeptics ... They're actually eroding trust in science and the scientific community
(Gopal, 2023)", while another had the title "Huckabee's climate-denial book targeted at children (Fisher, 2023)". A request to EverBright for permission to post the cover cartoon of "Kids Guide to The Truth about Climate Change" in this review on climate education was declined.





### 3.4.3.2.3 Heartland Institute

The conservative think tank Heartland Institute states what they do: "We focus on issues in education, environmental
protection, health care, budgets and taxes, and Stopping Socialism (2024)". In 2009, a booklet from Heartland, "The
Skeptic's Handbook", challenged "conventional wisdom" on global warming and was sent to 14,000 public school board
presidents (Taylor, 2009). The contents were debunked by a series of posts in DeSmog (Jacquot, 2008).

Six years later, the so-called Nongovernmental International Panel on Climate Change (NGIPCC) prepared *Why Scientists
Disagree about Global Warming* (Idso et al., 2015). The book dismissed the scientific consensus on climate change and
accused the IPCC of not being a credible source. The Heartland Institute distributed 300,000 copies of the book to K-12 and
college science teachers in the U.S. (Heartland Institute, 2015). This elicited an angry response from the Executive Director
of the National Science Teachers Association warning its members of the propaganda (see Bast, 2017), and a complaint by
the NCSE about Heartland forcing its "climate change denial literature on science teachers (Branch, 2017; see also Lee and
Banerjee, 2017; McKenna, 2017, 2018; Climate Town, 2023)". Senator Sheldon Whitehouse described the climate denial
propaganda by the Heartland Institute on the floor of the Senate:

> ... I would like to explore the Heartland Institute's latest gambit, which is to airdrop climate denial propaganda
> directly into children's classrooms.
>
> This spring, Heartland delivered packages to hundreds of thousands of K–12 and college-level science teachers
> across the country. These materials were designed to have a veneer of credibility. Each one was stamped with the
> headline 'Study: Science Teachers Giving Unbalanced Education on Climate Change.' This intriguing story was
> attributed to something called Environment & Climate News ... It turns out that the Environment & Climate News
> is not actually news. It is not a news outlet. It is the monthly newsletter of, guess who, the Heartland Institute. They
> are citing themselves, masquerading their newsletter as a news outlet ...
>
> What we don't need are fossil fuel front groups pumping out more phony science to pollute public education, just
> like they pollute our oceans and atmosphere (2017).

Another assault against climate education from the Heartland Institute was the book – *Climate at a Glance for Teachers and
Students* (Watts et al., 2022) – mailed out to 8,000 middle and high school teachers in early 2022. The banner for the release
of the book read that the: "Book intended to be 'supplemental' to standard curricula and counter alarmist narrative with facts
on the climate that reflect current data and research (Heartland, 2023)". Furthermore, its purpose was presented as:

> Climate At A Glance puts frequently argued climate issues into short, concise, summaries that provide the most
> important, accurate, powerful information. The summaries are designed to provide a library of solid yet simple





rebuttals so that legislators, teachers, students, and laymen can easily refute the exaggerations of the so-called 'climate crisis' (Climate at a Glance, undated).

In their webpage announcing the release of *Climate at a Glance*, Heartland proudly publicized that: "The Economist magazine called Heartland 'the world's most prominent think-tank promoting skepticism about man-made climate change' (Heartland, 2023)".

The Heartland Institute, EverBright Media and CO2 Coalition are typical conservative climate-denial organizations.

### 3.4.3.2.4a Ohio Natural Energy Institute

The Ohio Natural Energy Institute: "is dedicated to educating people about the indispensable industry that makes life better for every Ohioan (2023a)", and "natural energy" to them is natural gas and oil. Their website has sections for both students and teachers, offering a variety of one-page pamphlets praising the fossil fuel industry. Teacher workshops have been attended by 3,310 educators from 1,632 schools in the state (Ohio Natural Energy Institute, 2023b).

No mention of climate change or global warming was found on their website, but their homepage did boast of a 37% reduction in emissions without acknowledging why such emission reductions were important. A separate pamphlet explains:
"EIA data shows that between 2005 and 2015 the Ohio power sector reduced their carbon emissions by 50 million metric tons a year (37.7%). This was the biggest reduction of any state in the country and largely due to the increased use of natural gas to produce power (Ohio Natural Energy Institute, undated)"; in other words, a fossil fuel saved them from a fossil fuel. The group has been accused of indoctrinating children with the benefits of fossil fuels while ignoring how fossil fuels contribute to climate change (Zou, 2017).

### 3.4.3.2.4b Ohio Higher Education Enhancement Act

On May 17, 2023, the state senate passed the "Enact Ohio Higher Education Enhancement Act" which declared the following:

"Controversial belief or policy" means any belief or policy that is the subject of political controversy, including issues such as climate policies ... faculty and staff shall allow and encourage students to reach their own conclusions
about all controversial beliefs or policies ... it will not endorse or oppose, as an institution, any controversial belief or policy ... Prohibit political and ideological litmus tests in all hiring, promotion, and admissions decisions, including diversity statements and any other requirement that applicants describe their commitment to a specified concept, specified ideology, or any other ideology, principle, concept, or formulation that requires commitment to any controversial belief or policy ... (Cirino, 2023, 25-28).



The proposed legislation would leave climate change open to debate for students to decide and, in effect, would restrict the hiring and promotion of teachers who support the science of climate change. The bill currently resides within the state House, where it currently does not have the votes to pass (Gearino, 2023).

While not dealing with primary or secondary education, but colleges and universities, the state bill, if passed, will be a major setback for climate education in Ohio. The bill had been sponsored by Senator Jerry C. Cirino who stated that "both sides of

the equation need to be understood" (Kowalski, 2023). "Both sides" means the introduction of climate-denial arguments (Kowalski, 2023; Waldman, 2023c).

### 3.4.3.2.5 Oklahoma Energy Resources Board

The Oklahoma Energy Resources Board (OERB), a state agency funded by the oil and gas industry, offers a "HomeRoom" for students and teachers. As with the Ohio Natural Energy Institute, they boast of reducing $CO_2$ emissions:

Thanks to clean-burning natural gas, the United States has reduced CO2 emissions by 14 percent since 2006, making us the envy of the industrialized world. No one else comes close. In the meantime, Oklahoma has done even better in the power sector and will continue to take measures to reduce all greenhouse gases (Oklahoma Energy Resources Board, 2024).

An issue with OERB's climate education is their cartoon mascot "Petro Pete", who especially appears in their audio-book

series, "Petro Pete's Adventure" (2022a), and in their curriculum called "Little Bits" for K to 2nd grade students (2022b) and "Fossils to Fuel 2" for 3rd to 6th grade students (2022c). Workshops for teachers are offered at no cost, plus a stipend of $100 to teachers. The Center of Public Integrity has pointed out that: "Oklahoma remains the epicenter of oil-industry puffery in the classroom (Zou, 2017; see also Wertz, 2017 Atkin, 2020; Tannock, 2020; Climate Town, 2023)". A request to OERB for permission to post the cover cartoon of "Petro Pete's Adventure" in this review on climate education received no response.

Both the OERB and the Ohio Natural Energy Institute are petro-pedagogy organizations.

### 3.4.3.2.6 PragerU

Prager University, well-known for its climate-denial views (DeSmog, undated c), is a conservative media outlet and not an accredited academic institution. However, in the summer of 2023, PragerU was approved as an education vendor in Florida. An article in *Scientific American* criticized the adoption of the PragerU material by the state in an article titled "DeSantis's

Florida Approves Climate-Denial Videos in Schools (Waldman, 2023b; see also Waxman, 2023)". Other states soon followed Florida in approving the PragerU programs, including Montana, New Hampshire, Oklahoma, and Texas (PragerU, 2024a).





An example of PragerU's educational material is an article in their "PragerU Educational Magazine for Kids" named "Ania's Energy Crisis". In the article, climate change is presented as an "unproven and debated" theory (PragerU, 2024b, 6). While

Ania supports action to phase out fossil fuels, her parents want her to hear "the other side (PragerU, 2024b, 12-13)". Her father, who "reads scientific journals and talks with researchers at his university", shares a series of standard climate-denial talking points with his daughter, and her mother also chimes in with more climate-denial points. Finally, the young girl wonders: "Was it possible that she had only been taught one side of the story?" Ania is quickly learning the denial lesson to doubt the science of climate change, and so will other children who read this story.

The political orientation of PragerU is clearly laid out in the preamble for their on-line petition to get PragerU into all American schools with its emphasis on a conservative agenda:

> PragerU is trying to help America's students—but the left (which has hijacked and controlled the education system, including teachers unions) is doing everything in its power to label us as 'far right,' deplatform us, and keep PragerU out of schools. The left wants you to think that the reason students are failing is because bureaucrats need
more money. The left wants you to believe that teachers unions are protecting kids and doing right by teachers. The left claims kids don't need patriotic education (2024c).

In an interview by *Time*, PragerU CEO Marissa Streit added: "America's education system has been hijacked by one side ... How are we going to have great teachers, if the teachers themselves are basically held hostage to one ideology? That is essentially what we're trying to break here (Waxman, 2023)". A request to PragerU for permission to post the cover cartoon

of "Ania's Energy Crisis" in this review on climate education received no response.

### 3.4.3.2.7 STEM Careers Coalition of Discovery Education

The STEM Careers Coalition of Discovery Education lists the American Petroleum Institute and Chevron as content partners: "Content Partners will support expanded student impact while sharing existing inspirational and high-quality content and provide subject matter expertise … (Discovery Education, 2024a; see also Winkel, 2022)". The goal of the

STEM Careers Coalition is to reach ten million teachers and students by 2025 (Discovery Education, 2024b). One of their STEM Careers Coalition programs for educators is called Effects of Petroleum on Our World, which makes no mention of climate change (Discovery Education, 2024c). A program was found on the impact of climate change on health (2024d). The STEM Careers Coalition is another petro-pedagogy organization.

### 3.4.3.2.8 Texas Education Agency

An in-depth exposé revealed the disruptive influence of the energy-industrial complex in Texas on climate education (Worth, 2022; see also Judy, 2023). The State Board of Education had a resolution that: "Instructional materials should present



positive aspects of the United States and Texas and its heritage"; on 2 February 2023, it was amended to read: "Instructional materials should present positive aspects of the United States and Texas and its heritage and abundant natural resources (Texas Education Agency, 2023, 9)". The added term "natural resources" is not defined, but clearly refers to oil and natural gas. There are concerns that science textbooks would have to promote fossil fuels and omit the threat of climate change to comply with the change. The amendment had been made by a State Board of Education member who justified the decision by stating: "Our schools are paid for by the fossil fuel industry for the most part ... (Waldman, 2023a)". In a separate move, the education board banned several climate textbooks in high schools because how the fossil fuel industry was being negatively portrayed (Salam, 2023).

For recent reports on climate denial and the influence of the energy-industrial complex on climate education in American classrooms see "The fossil fuel industry's public school takeover (Atkin, 2020)", "The Brainwashing of America's Children (Climate Town, 2023)", "How to Confront Climate Denial (Damico and Baildon, 2022)", "Cartoons, jokes, and lies: Kids at risk from climate change denial (Reid and Branch, 2023)", "What the latest assaults on science educations look like (Strauss, 2017)", "Texas Weakens Climate Science Education Guidelines", "DeSantis's Florida Approves Climate-Denial Videos in Schools", "Climate Science is Under Attack in Classrooms (Waldman, 2023 a-c)", "Miseducation, How Climate Change is Taught in America (Worth, 2021)", and "Oil's Pipeline to America's Schools (Zou, 2017)".

## 4 Discussion

The science of climate change has done just fine against the lies and propaganda of climate denialism and the science has only grown stronger over time. The problem has been that there is a glaring gap between scientific knowledge and public perception of that knowledge (and the scientific consensus). How did such a misunderstanding by the public take place? Climate-denial propaganda is the cause of this Consensus Gap. As the energy-industrial complex has poured millions of dollars into PR firms to promote its propaganda against the scientific consensus, climate denialism has crippled climate communication for decades, affecting climate education. Disturbingly, an unprecedented surge in climate denial by the energy-industrial complex and conservative organizations (and politicians) has recently fallen upon classrooms. This review is intended to be a call-to-arms before irreparable damage is done to the school system and knowledge building on the climate crisis.

Climate education, despite a serious and genuine effort, has failed to teach the world about the causes and risks of the climate crisis, and a major reason, if not the reason, is climate denial. An authoritative source of the scientific consensus on climate change is the assessment reports of the IPCC. However, the crucial anti-science role of climate denial needs to be recognized more precisely by the IPCC instead of scattered loosely throughout their reports. The science of climate denial is now much more important than the science of climate change itself, in terms of climate policymaking, but the IPCC does not





fully appreciate this. The IPCC has always had a report on the physical science of climate change, but policymakers and the public would be better served if the IPCC issued a separate report on the social science of climate denial in future assessments. Additionally, PR professionals could be hired by the IPCC to prepare and promote a "Summary for the Public"

to give climate communication a fighting chance and a "Summary for Kids" to aid in climate education.

Social scientists have already done a good job of identifying the sources, tactics, impact, and other aspects of climate denial, but in my opinion, have inadvertently portrayed climate denial as a legitimate response by conservatives seeking to uphold their worldview. This can appear to normalize their climate denial to the casual observer. A conservative bubble has been established to isolate themselves from the real world where lying, wilful ignorance, gaslighting and denying science are

allowed to protect the ideology, no matter the consequences. Such anti-social tactics of the propaganda campaigns used by conservatives have been under reported. Climate denial, by delaying necessary legislation on a global crisis, represents a form of deviant behaviour as lives are placed at risk so that certain conservative values are protected. Deviant behaviour associated with climate denial and its social movement are rarely explored by social scientists, with McKie being a notable exception in her article "Climate change counter movement organisations: an international deviant network (2022)".

Studies have recommended how to tackle climate denial. However, as discussed, nothing is working so far. Climate education has been relatively successful with liberals but has had no impact on conservatives in some countries for more than a decade. Without engaging this major sector of society, legislation to stop the climate crisis is impeded if not entirely blocked. In my opinion, something new is needed to limit the influence of climate denial on climate education. In *Climate Denial in American Politics: #ClimateBrawl*, I recommended climate brawl as a climate-denial countermovement:

780        Primarily, a climate brawl is a challenge to the hard-core climate deniers – the political elite, contrarians, and major Twitter [X] influencers – who are leading the spread of climate denial propaganda. The denial cabal and their contrarian allies must be forced to defend their false claims, which they cannot. The objective is to discredit climate deniers and their sources which, in turn, mutes the influence of the propaganda. The hard-core climate deniers are unlikely to change; so, they must be marginalized, and in the particular case of political climate deniers, they

785        become 'marginalized' by being voted out of office (Kutney, 2024, 243; see also 25-27).

This anti-denial strategy was developed during my decade-long challenge of hard-core climate deniers on Twitter (now X). The above quote was more in reference to political climate denial but should also apply to conservative climate-denial organizations and petro-pedagogy organizations. Social science research on the marginalization of hard-core climate deniers is recommended which would be complimented by additional study on the deviant and anti-social aspects of climate

denialism. The time has come for less study on why and how climate denialism exists and more study on why and how climate denialism should not exist.





Climate education has been an easy target for the energy-industrial complex, as our children, teachers, and schools are literally sitting ducks for their propaganda, especially when located in regions that are conservative and/or connected to the energy-industrial complex (more so when also underfunded by the government). Organizations in both camps are promoting

climate denial disguised as educational programs:

1) Conservative Organizations – the promotion of conservative values and denying the science of climate change (for example, CO2 Coalition, EverBright Media, Heartland Institute and PragerU).

2) Petro-pedagogy Organizations – the promotion of fossil fuels and ignoring the science of climate change (for example, Energising Futures, Energy4me, FortisBC Energy Champions, Inside Education, NXplorers, Ohio Natural

Energy Institute, Oklahoma Energy Resources Board, Scientix, STEM Careers Coalition and Switch Energy Alliance).

Future research on both types of climate denial organizations is recommended, especially (1) to explore the situation in countries not mentioned in this review, (2) to study the recent surge in organizations promoting conservative values, and (3) to identify other petro-pedagogy organizations.

Much of the world has little knowledge of the science of climate change and the scientific consensus. The lack of knowledge building is horrific after decades of climate science, climate communications, and climate knowledge. The energy-industrial complex and climate-denial organizations are to blame for this failure. They have turned their attention more to schools to ensure that it stays that way. Is climate denial in your school? Teachers, parents, and students should be looking. Raising awareness of the cagey practices of climate denial in public education will help identify and prevent it. Kids agree that there

is no room for climate denial in their classroom (Figure 3).




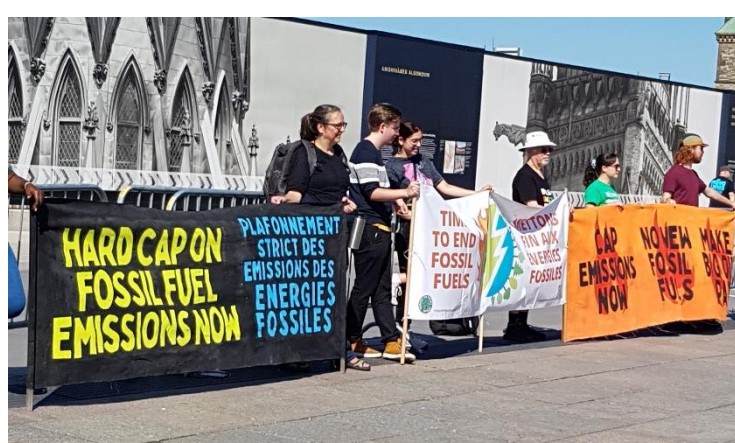
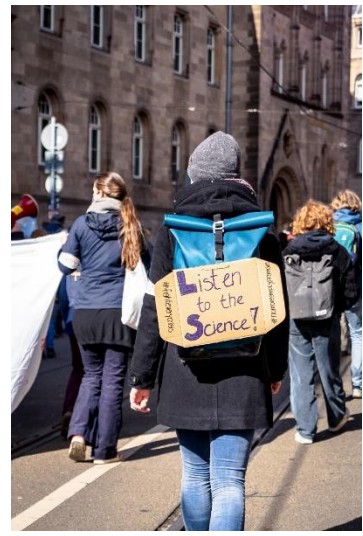

Figure 3. L – By permission of Gerald Kutney; R - Mike Baumeister on Unsplash.

## Competing Interests

The author is a climate activist on social media and has written op-eds in news media opposing climate denial and has a book on climate denial in American politics which has been used as a reference in this paper (Kutney, 2024). These have been
declared to identify any potential biases or self-promotion. The author has no other conflict of interests to declare.

## Acknowledgements

The author would like to thank David Crookall for his encouragement to prepare a manuscript for the special issue, and David, Solmaz Mohadjer, and Ellen Field for constructive criticisms and useful suggestions in its design and preparation.

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
