# Peer review of "Climate Denial - the Antithesis of Climate Education: A Review"

_EGUsphere, 2024_

## Author Comment (AC11)

Reviewer #1

GENERAL COMMENTS

The article seeks to provide a review of the threat of climate change denial to climate education, in particular formal K-12 climate science education, globally (in principle, but in practice with regard to a handful of mostly Anglophone nations). This is certainly a worthwhile goal. The article presents a lot of relevant information together, which is a definite plus. Not all of the information presented is correct, relevant, or contextualized, however, and it is presented with little expositional structure to aid readers' comprehension and no theoretical apparatus to extract any significant moral. That said, since the article is apparently intended for a special issue, it may be that it will well serve a purpose of that issue without substantial revisions. My formal ratings and recommendations do not take that possibility into consideration.

Addressing the list of questions provided by the journal (https://www.geoscience-communication.net/peer_review/review_criteria.html):

1. Does the paper address relevant scientific questions within the scope of GC?

Yes: the paper addresses the effects of climate change denial on climate education, and geoscience education is among GC's main subject areas.

2. Does the paper present novel concepts, ideas, tools, or data?

In general, no: as the subtitle indicates, however, it is intended as a review. The distinction between "petro-pedagogy" and "conservative" (better, "climate denial") organizations seeking to affect climate education is helpful and has not been explicitly made elsewhere, as far as I know. I have changed the phrase from "conservative" to "climate denial."

3. Are the scientific methods and assumptions valid and clearly outlined?

There is little application of any method evident. The brief method section is not detailed enough to enable replicability, and in any case the author seems to have just looked in the literature and on the web for relevant information, which isn't really a method. This is not necessarily objectionable in a review. I have expanded the methodology.

4. Are the results sufficient to support the interpretations and conclusions?

No specific results or conclusions are identified as such. The article successfully establishes that climate change deniers are attempting to undermine climate education in at least a number of developed countries -- as acknowledged, relevant information is scarce for the majority of nations. Information about the effect of these attempts is often not available, however, which is not acknowledged. Competing and complementary hypotheses about the comparative weakness of climate education (especially inertia in the U.S.) are not adequately examined. The author sometimes offers quantitative judgments without providing evidence (as noted in the specific comments). The author does not always clearly distinguish between the goals of climate change deniers with regard to education and their successes in accomplishing those goals (as noted in the specific comments). Excellent point, I have stressed that there has been little study on the success of these goals of climate denial.

5. Do the authors give proper credit to related work and clearly indicate their own new/original contribution?

The author scrupulously credits related work. Since the article is a review, there is no new/original contribution to speak of, except for the assembly of the information, the distinction between "petro-pedagogy" and "conservative" organizations, and the (undeveloped) suggestion that "climate brawling" might mitigate the effect of climate change denial on climate education.

6. Does the title clearly reflect the contents of the paper?

Yes, except insofar as it would be better to describe climate denial as "antithetical to" than as "the antithesis of" climate education. I am changed the title - Climate Denial and the Classroom: A Review

7. Does the abstract provide a concise and complete summary?

The abstract is concise. It focuses on "climate-denial" (elsewhere described as "conservative") organizations while apparently neglecting "petro-pedagogy" organizations, which are treated at a similar length in the manuscript; I would recommend referring also to the latter in the abstract. The abstract should also reflect the (understandable) restriction of the discussion to English-language sources and the nations primarily discussed (the U.S., Canada, the U.K., the Netherlands, etc.).

These additions should be possible without making the abstract too lengthy. I made the changes to the abstract.

8. Is the overall presentation well structured and clear?

No. There is a lot of material with a lot of details, especially in section 3 (24 manuscript pages). Not all the material and not all of the details are relevant to the main theme of the article, there were sections that seemed out of order or out of place altogether, and the inclusion or exclusion of examples often seemed arbitrary (see the specific comments for details). With no argumentative or expositional throughline provided, it was tiring to read the article closely. Adjusted as per your specific comments below.

9. Is the language fluent and precise?

Sentence by sentence, the language was reasonably fluent. There were a number of passages in which it was not precise (see the specific comments for details). Of particular concern is "climate denial," which (as is usual) covered a lot of ground, from denying that it's real through denying that it's us to denying that it's bad (to use Maibach's formulations). The author is clearly aware of the range here but wasn't always careful about specifying what form of climate denial was under discussion (as noted in the specific comments). In my opinion, there are many shades of climate denial but have an unwritten common goal – to delay any legislation to stop climate change, especially in regards to the use of fossil fuels.

10. Are the number and quality of references appropriate?

The references, which occupy pp. 31-45 of the manuscript, are abundant -- perhaps overabundant. Many of them are of low quality but for explicable reasons: the author dutifully cites a lot of climate change denial literature. (The journal might give thought to separating these into a reference list of their own.) It was surprising not to see more use of the science education literature's discussion of climate denial. If there are a few specific references that you have in mind, I would be pleased to look at them.

SPECIFIC COMMENTS

1 (title): Climate denial is not the antithesis of climate education, although it is antithetical to it: the antithesis of climate education would be climate miseducation (or, to coin a phrase, diseducation, to invoke the contrast between misinformation and disinformation). The title has been changed.

19-24: On the strength of the quotation, Tutu was calling specifically for climate justice education, while the author is apparently calling for stronger climate science education; this is a bit of a mismatch. It has been removed.

34: In the U.S. at least, there are problems with climate education independent of climate denial. Agreed.

42-45: This laundry list isn't especially helpful. (Sealioning, e.g., is a tactic taking its name from a 2014 web cartoon [https://wondermark.com/c/1062/]: how well-known is it?) Rather than define all the terms, a more general description might be in order. Removed.

52-57: These activists are engaged in different fights in different venues using different tactics, and the list is notably incomplete for no obvious reason. The reader is left with the impression that this is an offhand and opinion-driven list. The paragraph now begins with - Opposition to climate denial generally has arisen, but is only briefly mentioned in this review. Popular (and my favourite) examples of climate activism include:

59-62: (1) The previous paragraph described the activists as opposing climate denial; this paragraph describes them as trying to stop propaganda funded by the fossil fuel industry. The latter project is at best a subset of the former, since not all climate denial is so funded, and not all opposition takes the form of trying to ban (as opposed, e.g., to debunk or to inoculate against). (2) There are quantitative claims here -- "unabated"; "surge" -- for which no evidence is adduced. Wording was change to - Nevertheless, despite such praiseworthy efforts to stop it, climate denial and the propaganda funded by the energy-industrial complex have continued, especially in America.

63: (1) Cite the previous studies. (2) The segue to "the classroom" is a little rough, since the activists of 52-57 are not uniformly concerned with the classroom. Paragraph changed to - Climate denial in the classroom is the focus of this review; for recent reports in America (for example, where the largest number of such organizations were found) see Atkin, 2020; Climate Town, 2023; Damico and Baildon, 2022; Noor and Westervelt, 2023; Reid and Branch, 2023; Strauss, 2017; Waldman, 2023 a-c; Worth, 2021a; and Zou, 2017. This review provides a summary of the climate-denial organizations that are the leading offenders in manipulating climate education in schools. An important general goal of the review is to create awareness of the growing threat in the classroom,

so that teachers and parents can protect children in their schools from the anti-science influences of climate denial, and climate education researchers and instructors are aware of this menace. Is climate denial in your school, or your child's school, or in any local school?

65-66: Climate denial organizations such as the Heartland Institute are not the only relevant organizations here; fossil fuel industry groups such as the Oklahoma Energy Resources Board play a role in climate miseducation too. (This is acknowledged later but it should be clearly stated from the outset.) No specific organizations are listed at this stage.

66-68: But educators and parents aren't the only stakeholders here (and they're not the most likely readers of the journal); perhaps discuss the interests of higher education instructors in having climate-literate students as well? See above for new paragraph.

70-71: With regard to the general public, "communication" is a better term than "education"; insofar as education is aimed at the general public, it's typically in the context of what's called "informal education" (at, e.g., museums). Climate change in informal education is important, too, but the article focuses on formal education. "Communication" is now used.

72-77: This isn't described in enough detail as to ensure anything like replicability. The methodology was expanded -

A chief task of climate communication is the teaching of the main messages of the science of climate change to the general public and in all levels of education. This review focuses on the most vulnerable sector, the children in primary and secondary levels (K-12 in North America), of education.

The general methodology used in this review was similar to my book, which included:

> This political issue can be emotionally charged. Scholarly research, however, requires an impartial approach, and an examination of climate denialism, therefore, cannot exclude consideration of any positive aspects; in this study, which traces the evolution of climate denialism, none were found, which would not surprise the majority of physical scientists who study the climate. Social scientists would likewise generally agree but have identified psychological and sociological factors to account for the rise of the climate denialism movement. Academic studies, along with my decade-long Twitter experience, have been applied in this extensive study of climate denialism (Kutney, 2024, 4).

References in the peer-reviewed literature were sought on the influence of climate denial organizations and/or the fossil fuel industry in schools, especially those recently published (since 2021), with selected earlier references. A comprehensive summary of such organizations was a major purpose of this review to illustrate the scope of such organizations involved in climate denial in the classroom. Grey literature sources were added for quotes, critical commentary, and up-todate news media information. Websites for organizations associated with climate education and those for groups promoting climate denial in schools have also been utilized. Generally, the peer-reviewed literature was found using Google Scholar and the grey literature using Google; specific searches included: "petro-pedagogy," "climate denial, schools", "fossil fuel industry, schools", and "petroleum industry, schools", and the names of particular climate-denial organizations in schools listed in this review. Studies picked up by these searches were also examined for other relevant references. Mainly references in the English language were examined.

The term "climate denial" is defined as: "those who deny the accepted science that greenhouse gas emissions must be stopped as soon as possible, as climate change is a present-day threat, is getting worse, and is mainly caused by us (Kutney 2024, p. 17)" and also includes climate denial by omission when teaching about the fossil fuel industry, but neglecting that the burning of fossil fuels are the main contributor to the creation of climate change (especially relevant to petro-pedagogy). Climate change denial is abbreviated in this review to climate denial, as with related terms such as climate change communication to climate communication and climate change education to climate education.

This review sets out to answer a series of questions as follows:

•       What is the current state of public knowledge of the science of climate change? To answer this question, recent surveys of public awareness on important messages from the science on climate change were examined.

•       What is hindering the public from gaining knowledge about the science of climate change? Again, recent studies were favoured, but more historical information was also included.

•       What organizations are attempting to hinder climate education in schools? The peer-reviewed and grey literature supplied direct examples of such organizations. Specific examples of how these organizations operated were found by examining their websites and publications. Peer-reviewed and grey literature commentaries on these organizations were also examined. Greater focus was given to recent information.

In the last section ("Discussion"), conclusions, recommendations and suggestions for future research are offered. These are based mainly on the findings presented in the "Results," but also my decade-long experience challenging climate denial on Twitter (now X) and the research for my book Climate Denial in American Politics: #ClimateBrawl (Kutney, 2024).

79-91: It's distracting, and it steps on the punch line, to announce the results when describing the questions (especially in the third bullet). Good point, such comments were removed, see above.

82-85: As phrased, this begs the question. Given that the public's knowledge of the science of climate change is not as great as might be hoped for, there are any number of possible explanations, of which the two most salient are that they're not being educated appropriately and that they are being educated appropriately but something (which could indeed be climate change denial efforts funded or inspired by the fossil fuel industry) is blocking their uptake of that education. In fact, both are probably at play -- bear in mind that in general public surveys, a majority

of the respondents received their education a long time ago. Excellent point, which has been acknowledged in the manuscript.

82-83: It is not likely that public opinion surveys are capable of identifying climate denial as harmful to climate education. (Which is not to deny that they can find correlations between climate denial and opposition to climate education, but such opposition doesn't automatically translate into concrete harms.) My intent was to show that there is a problem with knowledge, especially, about the scientific consensus of climate change. The wording was changed to make that clear.

98-99: But of course surveys of the general public, which are usually of adults, only provide information on the past general state of climate education, in some cases the distant past (see comment on 82-85). Corrected, as stated above.

100-105: It isn't clear that respondents are in any kind of epistemic position to have a reasonable view on Pew's question, so it isn't clear what the responses really signify. It would be helpful to discuss a number of different questions probing the same general area (e.g., trust in climate scientists, perception of consensus among climate scientists) to see if the results converge. Paragraph changed to - Recent surveys (since 2021) have revealed an alarming lack of understanding of the science of climate change by the public. The polls are presented in order of the geographical scope of the survey, beginning with one country (the United States) and ending with a survey of most countries of the world. The polls often ask a series of questions; the most relevant to the consensus view of the science of climate change by the public was only discussed. According to the IPCC, an important aspect of this consensus is that: "Human activities, principally through emissions of greenhouse gases, have unequivocally caused global warming ... (IPCC, 2023, 4)".

110-111: As described, the Pew survey does not show this. What results, from the same or different surveys, show this? This paragraph was replaced by - The IPCC assessments, for example, demonstrate that the climate scientists know "very well" that climate change is occurring and the causes (2023, 4); the Pew survey shows that less than half of Americans are aware of the scientific consensus on climate change.

112: Not "climate change awareness" but perhaps "attitudes toward climate change" (encompassing whether it's real, it's us, and it's bad, to borrow Maibach's formulation). Attitudes was added.

114: "the poll" -- Which poll (what year)? 2023 as suggested in 126? "taken in December 2022" was added.

123: Given that the science doesn't of itself provide policy recommendations, alarmed and concerned arguably match the scientific consensus equally well -- the only difference given is with regard to motivation. I debated this point myself. Science has promoted "urgency" to take action on climate change which "concerned" do not have. In any case, this is debatable, so I have added the "concerned" category, as you recommended.

125: "not very reassuring" -- To whom? About what? What figure would be reassuring and why? Paragraph now reads - Interviews (with 32 adults) were also held by the Pew Research Center with Americans "most skeptical about climate change" (the purpose of this question was to find out why some Americans did not see an urgency to deal with climate change, while scientists were calling for immediate action). The replies from "those most skeptical about climate change" provided their personal reasons for their climate denial:

128: Why is this population of especial interest (to Pew and to the project of the article)? See above

134: True, but (1) why is this of interest here? and (2) it suggests that they are mindless parrots in the thrall of industry shills, etc. -- which may be true, of course, but for all that has been said, it may be true that they're all fiercely independent thinkers who did their own research. Inferring from the uniformity of responses of a similar survey of the Alarmed that they were all mindlessly sharing common climate alarmist talking points would be objectionable, would it not? This sentence was removed.

140, 142-143: It might be helpful to report the general public figures together in one paragraph and then discuss the evidence for political polarization in another. Paragraph was change to - The Yale survey above had 53% of those surveyed (in the alarmed and concerned categories) agreeing that climate change was caused by human activity. The Pew Research Center also conducted a poll on public awareness of climate change in the fall of 2023. Only 46% replied that a "great deal" of human activity contributed to climate change, 19% among Republicans and 71% among Democrats (Tyson and Kennedy, 2023; see also Tyson, Funk, and Kennedy, 2023; Energy Policy Institute at the University of Chicago, 2023; Alvarez, Debnath, and Ebanks, 2023). Only about half of Americans in this group of polls were aware that modern climate change was caused by us, despite that this is unequivocal according to the IPCC in their latest assessment of the science of climate change (2023, 4).

144-145: (1) "Distrust" rather than "Trust"; (2) how was trust measured in this survey (which was Alvarez et al.?)?; (3) How did trust and political position interact in this survey (or was it not reported? I don't see it from a quick glance); (4) rather than mentioning the relevance of this survey

to education, it might be helpful to devote a paragraph to discussing the relevance of all of the cited survey research to education. See above.

155-156: The gloss is not accurate, since the authors are not actually discussing climate education (even though they should have been). Nor are they explicitly attributing the misconception that scientists disagree about climate change and the consensus gap to climate denial. The sentence was removed.

157-173, 179-183, 201-202: A lot of these results, though interesting, are not clearly on target, since they don't reveal anything about climate change denial or climate change education specifically. I agree. They have been removed.

173-178: This is out of place in a discussion of polling. I have placed in later in the manuscript, where it is more relevant.

183-184: It isn't plausible that the polling was able to identify causal factors, and the quoted sentence is not presented in the original as a result of the polling. (No citation is given in the original, presumably on the grounds that it's obvious to anybody paying attention, which is fair.) Removed.

195: Of "Alarmed" and "Dismissive," "Alarmed" best represents the scientific consensus, but see comment on 123. How would matters look if "Alarmed" and "Concerned" were taken together? As above, "concerned" were added to the "alarmed" stats.

197: See comment on 123. This paragraph was shortened to - "Any category below "alarmed" + "concerned" indicates some degree of climate denial, with the most extreme climate deniers found in the category of "dismissive" (only the United States was greater than 10% in this category)."

203-209: If the point of this section is to show evidence for the consensus gap and to argue for its importance (especially with regard to education), as this paragraph suggests, then a lot of the poll results that have been reviewed are not relevant. Such polls were removed.

208: Not all climate action requires legislation: consider executive, judicial, and administrative actions. Administrative actions are particularly important in the context of U.S. education, where (e.g.) state science standards and their implementation are generally in the hands of administrative

agencies (although often with executive and/or legislative control and oversight in the background). Other levels of government can help. However, executive orders are at the whims of the president, as were seen when Trump replaced Obama. The judicial, at least the supreme court of the U.S., has generally ruled that such policies/laws are the job of Congress. In my opinion, if the world has any chance of meeting the Paris targets, the U.S. Congress must be involved. This interesting topic is beyond the scope of this review. If you prefer, I can remove the mention of legislation.

222: The concern isn't supported by the poll results. Perhaps the thought is that the poll results show that the concern is not being met? I replaced "concern" with "message."

226-230: This wanders away from the theme that there is a global call for climate change education in the direction of the theme that the call has not been adequately answered; restructure for clarity. I changed the last two sentences to - The UNESCO survey results indicated that most countries were not meeting their international treaty obligations on climate education; the overall finding was that 93% of countries had no or a very minimal level of content on climate change in their national curriculum (UNESCO, 2021, 12). Another UNESCO study found that:

238-240: But this is about climate change communication, not climate education. Your comment and similar ones are most appreciated. I found myself blurring climate education, climate education, and public awareness of climate change as almost being synonymous terms which was a misunderstanding on my part. These comments apply to sections 3.2 and 3.3. Thank you again for pointing this out.

241-243: Are the following two quoted statements the whole of the "extensive[]" comments? They seem well short of extensive: more like parenthetical. Paragraph changed to - In the latest reports of the IPCC (AR6), two statements on climate education survived the gruelling approval process in the Summary for Policymakers (SPM) (i.e., only the most salient points appear in the SPM); first, in Impacts, Adaptation and Vulnerability:

252-259: It would be more helpful to briefly summarize what is said rather than list and cite the keywords. For one thing, the list doesn't provide clear evidence that climate education specifically is discussed, only that climate denial is. You are correct about climate education; I removed the phrase climate education. I prefer to leave these various terms in, as some may not directly connect these phrases to climate denial.

265-266: But only briefly and often only by implication; the AR6 certainly could have done better. Sentence changed to -  In summary, the AR6 reports of the IPCC acknowledge how climate

communications, awareness, and action have been adversely impacted by the constant propaganda of climate denial that has deceived the public.

276-277: This is a non sequitur. It would only be plausible if the fossil fuel industry substantially affected the education recommendations of COP28 and the like and if national and subnational education authorities paid attention to those recommendations. No evidence for either has been presented (and it's doubtful that there is any). Fair comment, I have removed the last sentence.

290-291: This suggests that the previous paragraph described journalism about climate denial affecting climate education specifically, which it didn't. "Climate education" was removed.

293-299: This whole paragraph might be aptly moved to be the second paragraph of sec. 3.3. Done

304-306: Despite its use of the term "denial," it's doubtful that Freudian psychology has anything important to contribute to the modern discussion. Removed.

304-313: The structure of the exposition here is not clear -- hopping from the history of the concept of denial to a brief mention of climate denial in sociology to a comparison of two psychological studies of climate denial. Restructure for clarity. Paragraph changed to - An early report specifically on the psychology of climate denial appeared in 2001. The study examined "socio-psychological denial mechanisms". The authors found that: "The most powerful zone for denial was the perceived unwillingness to abandon what appeared as personal comfort and lifestyle-selected consumption and behaviour in the name of climate change mitigation (Stoll-Kleemann, O'Riordan, and Jaeger, 2001, 113)". In a second study two decades later, a dichotomy had arisen. The majority accepted the serious nature of climate change and supported "mitigation in the abstract", but were still not doing much individually, waiting for others to act first (Stoll-Kleemann and O'Riordan, 2020, 12; see also Bushell et al., 2017; Wullenkord and Reese, 2021; Berkebile-Weinberg et al., 2024).

314-322: The cognitive biases mentioned in the first sentence are not listed here. Prebunking is not defined. Gornsey and Lewandowsky 2022 and Jenny and Betsch 2022 are not adequately described. Section was expanded - Several cognitive biases in climate denial (attentional, perceptual, recall, confirmation, present, status quo, pseudo inefficiency, single-action) and ways of overcoming them have been reviewed (Zhao and Luo, 2021). Psychologists have found that prebunking ("first, an explicit warning of an impending disinformation attempt and, second, a refutation of an anticipated argument that exposes its fallacy") is more effective than debunking in the case of the climate propaganda (Lewandowsky, 2021, 11-12; also see Lewandowsky et al., 2020). In November 2022, a collaborative publication between the journals Nature Human Behaviour and Nature Climate Change was called "Climate change and human behaviour"

(Antusch and Yan, 2022). In this series, an article "A toolkit for understanding and addressing climate scepticism" by Hornsey and Lewandowsky (2022) examined the psychological origins of climate denial and how to challenge it (see also Wong-Parodi and Feygina, 2020; Ekberg et al., 2023). The abstract for the paper began with:

Despite over 50 years of messaging about the reality of human-caused climate change, substantial portions of the population remain sceptical. Furthermore, many sceptics remain unmoved by standard science communication strategies, such as myth busting and evidence building. To understand this, we examine psychological and structural reasons why climate change misinformation is prevalent (Hornsey and Lewandowsky, 2022).

They examined the interplay between personal and organized drivers of climate denial in Europe and on a global scale. In Europe, the denial machine was delaying action to stop climate change. The following strategies were presented for reducing the damage of climate denial:

1)      "appealing to sceptics through value-based frames"

2)      "appealing to sceptics through co-benefits";

3)      "leveraging climate-friendly actors within the conservative movement";

4)      "establishing norms";

5)      "consensus messaging";

6)      "embedding climate-friendly actions in social practice (Hornsey and Lewandowsky, 2022)".

For the degree of climate denial outside of America, see also Dunlap and McCright, 2015, 318-320; McCright, 2016, 79-82; Eichhorn, Moltof, and Nicke, 2020, 16, 45; Nartova-Bochaver et al., 2022; McKie, 2022, 2023; Berkebile-Weinberg et al., 2024, Vowles, 2024.

Also, in this special Nature issue, an article by Jenny and Betsch was titled "Large-scale behavioural data are key to climate policy", where they explained that: "improving individual knowledge through better communication alone is insufficient (2022, 1444)" and that industries must be targeted to deal with the climate crisis. Their main message was that not enough attention was being paid to behaviour science and concluded: "In addition to the structures that allow data collection, we urge governments to install structures that foster exchanges between scientists, politicians and the administrations to finally facilitate the actual use of behavioural evidence (Jenny and Betsch, 2022, 1447)".

323: Is fear a cognitive bias? If not, how is the claim that climate denial arises out of fear (and fear alone?) to be squared with the psychological literature that climate denial arises out of certain (here unnamed; see comment on 314-322) psychological biases? Sentence removed.

331-333: Only one of these arguments (about the national economy) seems relevant here, as reflecting conservative fears of climate change; why not stress it (perhaps to the exclusion of the other)? I have found that both are common "excuses" to justify not acting on climate change (the

goal of climate denial), especially among conservatives. Both are common to this day in social media and Congressional hearings.

334-335: What is "a force of anti-reflexivity" supposed to mean? Explain or omit. Deleted.

335-338: This is no help with regard to "anti-reflexivity." The end of the paragraph has been amended to - In 2010, McCright and Dunlap reviewed again the opposition of conservatives to environmental issues as a countermovement to defend the "industrialist social order (2010, 104)". Later McCright wrote that: "the climate change denial countermovement as a collective force defending the industrial capitalist system (2016, 77; see also Brulle, 2014; Dunlap and McCright, 2015)". Recently, Ruth McKie concluded:

> The USA was the birthplace of the organized opposition that emerged to challenge environmental legislation and climate action. Its roots stemmed from the purposeful consolidation of an action plan by the fossil fuel industry and vested interests ... If it were not for this organized campaign, countermovement opposition organizations across other countries may not have had the opportunity to emerge and garner success (2023, 43).

339: There's a gap between the claim and the Lewandowsky report, which can be filled by the thought that (formal) education is the best, the most significant, the most likely ... way of acquiring knowledge about climate change. Also: matters in what way? to what? Paragraph changed to - Barriers stand in the way of public knowledge about climate change, including climate denial, which is serving vested interests, such as hard-line conservatives and the energy-industrial complex (Oreskes and Conway, 2010, Chapter 6; McCright and Dunlap, 2011; Dunlap and McCright, 2015; Hornsey, Harris, and Fielding, 2018; Lewandowsky, 2021; Hornsey and Lewandowsky, 2022; Kutney, 2024). Stephan Lewandowksy has warned: "These political implications have created an environment of rhetorical adversity in which disinformation abounds, thus compounding the challenges for climate communicators (2021, 1)", and "The terrain for climate communications is treacherous (2021, 8)" because of the adversarial environment created by climate denial. Stoetzer and Zimmermann reached similar conclusions:

340: Insofar as "namely" suggests that the only such barriers are those presented by climate denial that is serving vested interests (which, presumably, is not synonymous with climate denial tout court), this seems wrong. Inertia throws up barriers of its own. See 339

343-345: This is about communication; the connection to education needs to be forged. See 339

346-349: Here too the connection to education needs to be forged.  See 339

350-352: The cited studies establish that climate change education is ineffective among conservatives, not that climate change education is being impeded by conservatives, so this is just off point. Sentence removed.

352-353: This premise has not been established by the preceding discussion (although it is by the following discussion). Climate education changed to climate communication.

353: See comment on 1. Changed as in 1.

359: What evidence is there for the comparative judgment ("more dire")? More dire in what aspects? In some ways, the situation is less dire, at least in the U.S., as shown by better treatment of climate change in state science standards and better preparation of science teachers between 2012 and the present. Sentence changed to - Over a decade later, the campaigns promoting climate denial in the classroom have escalated, as discussed in this section.

361-372: "softer" in what respect? It's clear enough that the most prevalent forms of climate change denial have been softening -- moving away from denying "it's real" and "it's us" and toward denying "it's bad" (to use Maibach's formulation) -- in general, and there's evidence that this is true of climate change denial campaigns targeting K-12 science education in the U.S. But the groundwork hasn't been laid in this article to discuss this transition here. This paragraph is also hard to follow in the absence of concrete examples. First sentence changed to - Climate denial in the classroom includes "petro-pedagogy". The term had been used to describe the energy-industrial complex funding energy and climate education programs for K-12 education, especially in STEM (science, technology, engineering, and mathematics) education (Eaton and Day, 2020, 462).

364: Likewise, "traditional conservative climate denial" hasn't been defined, so the contrast here will not be understood except by a reader already familiar with the situation. See above.

364: The groundwork hasn't been laid for the idea that the energy-industrial complex is providing K-12 science educational content (which moreover needs to be distinguished from the content provided by climate change denial organizations such as the Heartland Institute: see comment on 65-66). See above.

374: There are different relations being obscured by the word "sponsor" here: for example, actual fossil fuel companies provide the budget of the Oklahoma Energy Resources Board, but the

Heartland Institute hasn't received directly traceable funds from such companies for a long time now, and many of its corporate sponsors made a point of cutting ties with them after the Unabomber billboard fiasco. It's misleading not to distinguish the different relations. Last paragraph changed to - Below is a description of climate-denial organizations and petro-pedagogy organizations promoting climate-denial in the classroom (also see figure 2 for editorial cartoons reflecting conservative climate denial views and liberal views of petro-pedagogy).

374: The parenthetical description doesn't capture the fact that the lefthand cartoon in figure 2 reflects conservative views of climate education while the righthand cartoon reflects liberal views of "petro-pedagogical" efforts. See above.

380-393: (1) What evidence is there that this program is global in actual reach? (Anybody can put material on the internet; that doesn't mean that it's global.) (2) This is under the rubric "Climate Denial in the Classroom" (the title of sec. 3.4); from the description this program is at worst soft denial. Does it make sense to begin or to spend a lot of time with the less pernicious examples? I placed groups globally if that is what they claimed. This group does fit the definition of petro-pedagogy.

394-409: The same comments apply as for 380-393. Also, is there any information on the content of Switch Classroom in particular? I agree that both groups appear to be less known, but they are trying. I wanted to flag as many groups as possible.

430-442: Is there any information on the impact on EU science classrooms? You made this good point above, on what is their impact; and that is a question that I found no answer, but they are a threat as they promote propaganda in schools.

444-472: Is there any information on current impacts? From what's said here, it sounds like these efforts wrapped up about ten years ago (unless the festival was later -- it's hard to tell; it may be a yearly event). Not recently, but I did add dates to the statements.

475-481: This section seems to be trying both to introduce the Canada material and to discuss the Saskatchewan episode; it would be clearer if the latter were moved into its own section (and if the dependence of the Canadian economy on extraction industries was discussed further). This paragraph was intended to be more an introduction to petro-pedagogy and was moved to section 3.4 as an introduction (with suitable modifications).

483-494: Interesting but not a lot of details about the actual content or uptake. Some websites did not provide many details, unless you signed up as a student or a parent.

495-502: The same comments apply as for 483-494. And if the worst aspect is that it promotes only personal action on climate, it's fairly soft denial. Soft yes, but a standard ploy of the fossil fuel industry to place the onus about the climate crisis on us and not them.

503-506: There's so little information here that it's hardly worth including. I wish to include them just to warn readers of this new group.

507-511: The same comment applies as for 503-506. Same as above.

531-534: This shouldn't be in the Ten Peaks section: either in the introductory Canada section or a concluding section on its own. (Or perhaps the blank line 529 is supposed to set it off? Fair if so.) Yes, an extra line was added to separate it, as a short summary.

536-541: (1) Yale has had a series of surveys with this question, with a bit of up and down but generally in the mid-to-high 70s. (2) Other surveys have addressed the issue with different questions, confirming the high level of support but offering further insights that may be worth discussing here, e.g.:

Kamenetz A. 2019, April 22. Most teachers don't teach climate change; 4 in 5 parents wish that they did. National Public Radio.

Pizmony-Levy O, Pallas, A. 2019. Americans endorse climate change education. Teachers College, Columbia University. https://www.tc.columbia.edu/media/centers-amp-labs/the-public-matters/AMERICANS-ENDORSE-CLIMATE-CHANGE-EDUCATION-final-version-posted-v09172019.pdf

Lange J. 2023, December 15. Poll: Americans overwhelmingly want climate change taught in schools. Heatmap. https://heatmap.news/climate/education-climate-trust-teachers-poll#

Your sharing of the references was most kind. I only added the last one, as it was most recent. I was not familiar with Heatmap; it is a nice site.

542-545: There have been multiple attempts to introduce the CCEA in both houses of Congress, most recently S. 4117 and H.R. 7946 (both introduced after the paper was submitted). Reference added, thanks for sharing.

546: Public education in the U.S. is ultimately controlled by the state, but the bulk of decisions on curriculum and instruction are made at the district level -- and there are about 13,500 local school districts -- or below (school, department, classroom). So there's even more decentralization than is revealed here. Role of district level was added.

546-548: (1) This is irrelevant to the decentralization point. (2) At face value, the campaign was deeply misconceived, not taking into consideration the facts that standards are revised on a multiyear schedule and that education policymakers are not likely to be responsive to single-shot petitions. (Of course, the ulterior motive may have been just to harvest addresses from people concerned about climate change education, and in this it may have succeeded.) Interesting, but I left it in to demonstrate that some groups were attempting to promote climate education at the state level.

550: 26 states were involved with (were "Lead State Partners" on) the development of the NGSS; not all adopted them (as correctly implied below). Added

557: the 24 (actually 25 now) states said to be "using [the NGSS] as guides" have actually based their standards on the same National Research Council Framework on which the NGSS are based -- not much difference in practice, but it makes a difference in some contexts. Comment on Pennsylvania added.

557-558: since 2020, PA moved from the non-Framework category to the Framework category, so only five states are neither NGSS nor Framework: TX, FL, OH, VA, and NC. Reference added, thanks.

560: re "not guided by the NGSS," see comment on 557 The change of Pennsylvania was noted.

575-579: More details about what the study sought to understand, and more specific reportage on its results, are needed. The study looked for state-level (i.e. state board of education or state department of education) policies regarding climate change in four contexts -- "1) institutional governance, 2) teaching and learning, 3) facilities and operations, and 4) community partnerships" -- of which only 2 is really relevant here. The new paragraph reads - A study of climate education reviewed: "802 publicly available education policies across the United States" and "used a whole institution approach for data collection and analysis and considered four institutional domains of

potential climate change activity: 1) institutional governance, 2) teaching and learning, 3) facilities and operations, and 4) community partnerships (MECCE and NAAEE, 2022, 4-5)." Among their findings were that all states had policies mentioning climate change, but 33 states had very low focus, and 14 states had low focus on climate change content (9-11, 24, 40). States that followed the NGSS were more likely to include climate change content (9, 24-26). When energy was taught, there was little mention of climate change (9, 31-37). The report again highlighted the issue of climate denial: "For decades, political and social will to act on climate change was quickly swept away in a current of denial, avoidance, and political posturing (3; see also 7, 28, 44)".

592-607: It's surprising not to see any mention of the CO2 Coalition's recent attempt to disseminate its materials at a NSTA conference. See e.g. https://www.washingtonpost.com/climate-environment/2023/04/11/co2-coalition-climate-denial/ Thanks this reference. The following has been added - They describe themselves as: "comprised of more than 100 of the top experts in the world who are skeptical of a theoretical link between increasing CO2 and a pending climate crisis while embracing the positive aspects of modest warming and increasing CO2 (CO2 Coalition, 2023, 1). Among their activities is the development of programs on science education:

> In early 2021, a group of CO2 Coalition members decided to act on their concerns about the state of science education in America. They recognized that the teaching of science had strayed from the 400-plus-year-old scientific method and was less inclined to encourage inquisitiveness in students and more prone to require conformity to the opinions of teachers. At present, much of the instruction on climate change resembles an indoctrination into a political agenda rather than the provision of necessary tools for critical thinking (2023, 1).

On March 23, 2023, they issued a booklet attacking the position of the National Science Teaching Association on climate change (CO2 Coalition, 2023), using standard climate-denial talking points. The final conclusions of the booklet were:

> As a result, students are undergoing an indoctrination into a dangerous political agenda that ignores the enormous benefits of CO2 – a gas critical to life – and promotes an impossible objective of supporting modern economies without carbon-based energy sources.

> We respectfully urge the National Science Teaching Association to seriously consider a rejection of their previous endorsement of scientific censorship and return science education to the foundations of reason, open scientific debate and tolerance for alternative thinking (CO2 Coalition, 2023, 16).

The booklet was released as the National Science Teaching Association was holding a convention, where the CO2 Coalition had a booth and distributed the booklet and a comic book "Simon the Solar-Powered Cat," depicting carbon dioxide as being good for the planet. An article in the Washington Post about the episode warned that the CO2 Coalition literature could cause teachers to spread propaganda about the science of climate change to their students (Joselow, 2023). The members of the CO2 Coalition were kicked out on the second day of the convention.

612: "yet" is gratuitous Deleted

618-651: Is there any evidence of uptake by teachers? This has not been investigated to my knowledge.

652: It's hard to know how to assess this claim. They certainly have their differences (e.g., CO2 Coalition is single-issue and the others aren't; EverBright Media is for-profit and the others aren't; etc.). Agreed but the propaganda they all share is typical climate denial talking points.

652: Why no mention in this context of PragerU Kids, which isn't an outlier compared to these three and whose climate change denial videos and comics aimed at kids grabbed headlines in 2023 in a number of states, especially FL? See e.g. https://www.politico.com/news/2023/08/09/in-desantis-florida-schools-get-ok-for-climate-denial-videos-ee-00109466 PragerU was added

653-664: It would bear mention that ONEI is the OH equivalent of the OERB. Added to section on the OERB.

665-681: (1) The bill was amended to refer to climate policy rather than climate change, so it was transformed from hard to soft denial. (2) As acknowledged, this bill isn't about K-12 education, so why include it? (3) The bill hasn't passed or been enacted, so why include it, especially when a number of bills that actually sought to undermine K-12 climate change education in various states aren't discussed? Removed

682-695: It would bear mention that OERB was the first of its kind, with ONEI and others inspired by it. Why no mention of e.g. the Illinois Petroleum Resources Board? See e.g. https://www.levernews.com/a-fossil-fuel-miseducation/ Added -

3.4.3.2.4 Illinois Petroleum Resources Board

The goal of the Illinois Petroleum Resources Board is to "improve the image and credibility of the Illinois oil and gas industry", and this is accomplished through seven objectives, of which the first is "Education: Create an understanding of the Illinois oil and gas industry and good safety practices through programs with schools, organizations and the public at large (Illinois Petroleum Resources Board, undated a)". They offer a series of professional development programs for middle and high school teachers (Illinois Petroleum Resources Board, undated b). No connections between fossil fuels and climate change were found on their website. Blogs on their website generally defended

petroleum, including one titled "Benefits of Fossil Fuels to Humanity Have Far Outweighed Negatives":

> But it is dangerously misleading to focus exclusively on those [environmental] impacts and completely ignore their massive benefits. And using this deeply flawed framing as the basis of campaigns to rapidly eliminate the source of 83 percent of the world's energy and virtually all our modern products is even more dangerous considering the favored "alternatives" are completely inadequate to replace fossil fuels.

> It can't be emphasized enough that renewable energy – specifically wind and solar – can only generate electricity, and do so only when the sun is shining and the wind is blowing (Whitehead, 2022).

An article criticizing the Illinois Petroleum Resources Board was called "A Fossil Fuel Miseducation", which stated: "the IPRB doesn't appear to deny climate change — they mostly seem to avoid mentioning it at all. Instead, the group focuses on economic arguments about the oil and gas industry, which they claim will be a good source of jobs for decades to come, despite mounting evidence to the contrary (Gopal, 2024)".

698: Not "an education vendor"; rather, the DOE gave its imprimatur to the use of PragerU Kids videos in middle school social studies classes. Importantly, (1) it would be hard to justify the use of its climate change denying videos on those classes, given the lack of climate change content in the corresponding state standards, and (2) the DOE's imprimatur is basically irrelevant, since districts make decisions about instructional material, and some of the bigger districts have already said that they will not allow PragerU Kids materials to be used. Changed "an education vendor" to - allowed in Florida schools.

701: These states took different actions on different PragerU Kids products; it's misleading to lump them together as all approving the use of climate change denial material.

* In MT, the state superintendent of public instruction, Elsie Arntzen, a Republican, signed a textbook license agreement with PragerU. This doesn't have much actual significance, because the only requirement for obtaining such a license is posting a surety bond -- in PragerU's case, for $5000. Having the license doesn't mean that instructional materials will be considered, let alone approved, and the superintendent doesn't make decisions on instructional materials anyhow. But Arntzen expressed her approval of the materials, which might conceivably have some effect on districts' decisions.

* In NH, high school students in New Hampshire now have the opportunity to satisfy their financial literacy graduation requirement with PragerU Kids's Cash Course module online. The commissioner

of the state department of education, Frank Edelblut, a Republican, even appeared in a promotion for it. Whatever you think about this, it's not likely to have any effect on climate change education.

* In OK, the state Department of Education under state superintendent of public instruction Ryan Walters, a Republican, announced a "partnership" with PragerU, which seems to take the form of the department endorsing its videos for use in social studies classrooms. While there are opportunities to discuss climate change in social studies classes, Oklahoma's social studies standards don't provide a lot of opportunities for it, so the effect on climate change education will probably be limited. The Oklahoma Education Association reacted to Walters's announcement by reminding districts that they don't have to use the material and parents that they could opt their children out of exposure to them.

* In TX, a PragerU promotion that featured praise from Julie Pickren, a Republican member of the state board of education, claimed that PragerU is an approved education vendor in the state. That was not, and still is not, actually true. Texas has a lot of problems with climate change education, thanks in part to Pickren, but official approval of PragerU for Kids materials is not among them. I changed the wording to - other states expressed interest in the PragerU programs.

703-709: It's surprising not to see mentioned the not-very-hidden analogy Jews in the Warsaw Uprising:Nazis :: climate change deniers:climate change accepters. Not that I disagree, but I prefer to stay away from this comment, as it invites new criticisms from climate deniers.

726: It's actually a lesson plan for grades 3-5, and it has API branding. While it can be read as soft denial, it's intended to be a career lesson, not an environmental science lesson. Changed to - lesson plan for grades 3 to 5.

729: the heading should be "Texas State Board of Education" -- the TEA is basically the TX department of education; it's administrative and doesn't set policy and can't be blamed for the shenanigans discussed in this section Changed as recommended

729-739: (1) This is state action, so it doesn't seem like it belongs in this section. (2) There have been similar episodes elsewhere, with executive and legislative actions aimed at inhibiting climate change education and apparently motivated by climate change denial; why aren't they discussed as well? The Texas case has attracted much attention and has implications for publishers generally.

737-739: (1) This was not "unrelated"; the changes to the board operating rules were made in order to facilitate the later attacks on the textbooks. (2) The textbooks in question were not banned; they

were not approved. Districts are still free to use them if they wish; it's just harder and more expensive for them to do so. (3) The textbooks were not climate textbooks but grade 8 science textbooks. Last lines changed to - Textbook censorship in Texas, and other states, are increasing. One local school district in Houston voted to censor chapters on climate change; a local parent feared: "It's really kind of alarming what this could mean for ideological influence and control over what is taught in schools (Salam, 2024; see also 2023)".

740-746: This isn't part of section 3.4.3.2.8, and should be set off from it somehow. Shortened and made part of the section 1.

752-753: "climate denialism has crippled climate communication for decades, affecting climate education": if education isn't included as part of communication, as seems plausible, then what are the causal processes at work here? It is somewhat plausible that there are relevant causal processes (e.g., if newspapers mistakenly present climate denial as equally plausible as climate science, teachers may be swayed). But climate denialism also affects climate education directly, as the author has observed at length, and plausibly these effects are stronger, so they too should be mentioned here. The ending part of the sentence was changed to - climate denialism has crippled climate communication and has directly had negative influence on climate education.

753-754: "unprecedented" and "recently" aren't clearly true. Evolution has been the subject of similar controversy for a century, and the author discusses episodes of climate change denial going back more than a decade. (And it would be odd for them to go back much further, since such episodes are basically backlash to the inclusion of climate change in K-12 education, which is probably less than 20 years old in the US, with a surge after the release of the NGSS in 2013.) Corrected, as suggested. I added your commented on the NGSS later in the Discussion.

758: "if not the reason": indeed, definitely not the sole reason. In the U.S. especially, climate change is homeless because of a historic (more than century-long) neglect of earth and environmental sciences (as compared to biology, chemistry, and physics); even if climate change denial vanished overnight, there would remain a tremendous amount of inertia to overcome to get climate change taught adequately at the K-12 level. Sentence removed.

759-760: But likewise the crucial role of climate education, which is typically scanted in IPCC reports and the like (e.g., the NCA in the U.S.). Agreed

765: The "Summary for Kids" is a nice idea, but also why not a "Summary for Teachers"? Added

766-791: This all seems off-topic, with no visible connection to climate education (except for 787-788, which mechanically extends the recommendation of "climate brawling" to organizations working against climate education without providing any details about how this might work or have to be adapted, let alone any evidence of its effectiveness). Removed

794-801: This is a useful distinction. (1) It would be useful to place it earlier. (2) Thought might be given to noting the differences. "Petro-pedagogy" organizations tend to be less extreme and/or softer in their climate denial; they often have better connections with the world of education; they are probably more susceptible to public pressure (since they typically are themselves or are funded by publicly traded companies). "Conservative" -- or better "climate denial" -- organizations thus tend to be more extreme, less connected, and less susceptible. These differences make a difference in how to try to eliminate or mitigate their influence on public education, obviously. It is surprising that, having made the distinction, the author fails to extract these pretty obvious implications. I am not sure that I follow the "public pressure" comments. Most of these organizations are directly or indirectly sponsored by the same fossil fuel industry. The petro-pedagogy organizations are already embedded in the school system in some cases (BP being a notable example) which would make schools more addicted to the free "education" tools. As suggested, this section was placed earlier.

803: It's not obvious that there has been such a "recent surge" or, assuming that there has been, that it isn't just a blip without statistical significance. This paragraph was changed to -

Future research on both types of climate denial organizations is recommended, especially:

• to explore the situation in countries not mentioned in this review,

• to study if a recent surge in organizations promoting climate denial have developed,

• to determine (a) how effective these organization have been in getting climate denial into schools, (b) what damage has been caused and (c) what can be done to repair the damage..

809-810: (1) Is there evidence, such as polling data, to this effect? (2) The lefthand side of Figure 3 doesn't suggest that these students are specifically opposed to the inclusion of climate denial in their classrooms (though it's a safe bet). The righthand side doesn't either, but it's a bit closer. Removed the picture from the LHS.

819 onward: references, not reviewed

TECHNICAL CORRECTIONS all corrected (unless noted otherwise), thank you for this.

25: "Forward" should be "foreword"

28: citation should not be within quotation marks; this happens elsewhere (e.g., 136, 164, 185, 206, 262-263, 270, 271, 289, 294, 310, 321, 340, 384, 386, 389, 413, 429, 485, 520, 565-566, 568, 570, 573-574, 578-579, 582, 606, 610, 615 [twice], 628-629, 644, 665, 662, 693, 700, 705, 719, 724, 734, 737, 741-746 passim) and I have not necessarily caught every instance. I would normally agree but the guidelines for the manuscript indicated this format

29: "which shows that a problem exists" should be "revealing that a problem exists,"

55: "Canada's" should be set in romans Not sure what you mean; the official name is Canada's National Observer.

71-72: Something awry here -- delete "of education" or insert "sector" after "vulnerable" and add a comma after the right parenthesis.

81: "on" should be "of"

92: "negate" seems off: "mitigate"?

101: "A survey" should be "A series of surveys" given the time series mentioned in the next sentence.

110: "Kahn" should be "Kahan"

183: "effecting" should be "affecting"

244: "ity" should be set in italics

219: "Lewandowky" should be "Lewandowsky"

328: "fear of conservatives" should be "fear held by conservatives" or something like that, to avoid a misreading

329: does the author want to endorse the idea that individual freedom and free enterprise as understood by these organizations are indeed traditional American values, as this formulation seems to suggest, or to hedge here? Added interpretations of

350: "Kahn" should be "Kahan"

423: "their webpage" is ASE's page referring to Climate.Speaks? Yes, added ASE to make it clear

456: superscripted "2" thus in original? Mark with "[sic]" if so.

563: "2015" should be "2014-2015" (the survey received responses in late 2014 and early 2015)

580: book title should be capitalized

584: "Riley" should be "Rollie"

609: "Huchabee" should be "Huckabee"

623: It's called "NIPCC" not "NGIPCC"; clarify to what extent the NIPCC is the creature of Heartland added supported by the Heartland Institute

819 onward: hard to read without hanging indent! I agree but this was preset in the format for the manuscript

---

## Author Comment (AC12)

This paper aims to present a review of current knowledge on the influence of climate denial on climate education. This is a contentious but important issue that merits wider consideration and discussion within the geoscience education / communication community. There are issues with the paper as it's currently presented, however, and I recommend that these are addressed prior to final publication. In particular the focus on North America, almost to the exclusion of anywhere else, makes me question whether this paper should be re-cast specifically to focus on North America. I did limit my searches to English language articles, and most were found in the U.S.

1) Does the paper address relevant scientific questions within the scope of GC?

Yes. Climate denial and its influence on geoscience education falls clearly within the scope of the journal.

2) Does the paper present novel concepts, ideas, tools, or data?

This is a review paper so does not present new ideas emerging from novel research. It does make a contribution to synthesizing existing knowledge about climate denial in climate / geoscience education, but has some limitations in scope.

3) Are the scientific methods and assumptions valid and clearly outlined?

The methods used to identify, select, and analyse the information used in the review need to be more transparent. Granted this is a review article rather than a research paper, but this is a contentious subject and it's important that the paper does not lay itself open to accusations of using the 'obstructionist tools' associated with climate denial, e.g. cherry-picked data. Demonstrating a rigorous, systematic approach to information gathering should help to deflect such accusations. The recently-published systematic review of counteracting climate denial by Mendy et al. (2024): https://doi.org/10.1177/09636625231223425 provides an exemplar of the kind of approach appropriate for this type of review. The review article was included. The "Method" has been changed to -

A chief task of climate communication is the teaching of the main messages of the science of climate change to the general public and in all levels of education. This review focuses on the most vulnerable sector, the children in primary and secondary levels (K-12 in North America), of education.

The general methodology used in this review was similar to my book, which included:

> This political issue can be emotionally charged. Scholarly research, however, requires an impartial approach, and an examination of climate denialism, therefore, cannot exclude consideration of any positive aspects; in this study, which traces the evolution of climate denialism, none were found, which would not surprise the majority of physical scientists who study the climate. Social scientists would likewise generally agree but have identified psychological and sociological factors to account for the rise of the climate denialism movement. Academic studies, along with my decade-long Twitter experience, have been applied in this extensive study of climate denialism (Kutney, 2024, 4).

References in the peer-reviewed literature were sought on the influence of climate denial organizations and/or the fossil fuel industry in schools, especially those recently published (since 2021), with selected earlier references. A comprehensive summary of such organizations was a major purpose of this review to illustrate the scope of such organizations involved in climate denial in the classroom. Grey literature sources were added for quotes, critical commentary, and up-to-date news media information. Websites for organizations associated with climate education and those for groups promoting climate denial in schools have also been utilized. Generally, the peer-reviewed literature was found using Google Scholar and the grey literature using Google; specific searches included: "petro-pedagogy," "climate denial, schools", "fossil fuel industry, schools", and "petroleum industry, schools", and the names of particular climate-denial organizations in schools listed in this review. Studies picked up by these searches were also examined for other relevant references. Mainly references in the English language were examined.

The term "climate denial" is defined as: "those who deny the accepted science that greenhouse gas emissions must be stopped as soon as possible, as climate change is a present-day threat, is getting worse, and is mainly caused by us (Kutney 2024, p. 17)" and also includes climate denial by omission when teaching about the fossil fuel industry, but neglecting that the burning of fossil fuels are the main contributor to the creation of climate change (especially relevant to petro-pedagogy). Climate change denial is abbreviated in this review to climate denial, as with related terms such as climate change communication to climate communication and climate change education to climate education.

This review sets out to answer a series of questions as follows:

• What is the current state of public knowledge of the science of climate change? To answer this question, recent surveys of public awareness on important messages from the science on climate change were examined.

• What is hindering the public from gaining knowledge about the science of climate change? Again, recent studies were favoured, but more historical information was also included.

• What organizations are attempting to hinder climate education in schools? The peer-reviewed and grey literature supplied direct examples of such organizations. Specific examples of how these organizations operated were found by examining their websites and publications. Peer-reviewed and grey literature commentaries on these organizations were also examined. Greater focus was given to recent information.

In the last section ("Discussion"), conclusions, recommendations and suggestions for future research are offered. These are based mainly on the findings presented in the "Results," but also my decade-long experience challenging climate denial on Twitter (now X) and the research for my book Climate Denial in American Politics: #ClimateBrawl (Kutney, 2024).

4) Are the results sufficient to support the interpretations and conclusions?

A range of survey findings are used to imply the current state of public knowledge of the science of climate change, but I'm not convinced that all of the data presented are a valid representation of knowledge, or that this is a reliable indication of climate education. These data appear to relate to a range of constructs including awareness, perceptions, opinions and beliefs, which are not the same as knowledge. For example, having an awareness (being conscious) of climate change is not the same as having knowledge acquired through learning. It's also important to acknowledge that knowledge does not directly lead to actions. Rather than being 'ignorant of the irrefutable messages of the science of climate change and the scientific consensus' there is likely a much more complex interplay of factors – including education – influencing the gap between knowledge and behaviour (see review paper by Kollmuss & Agyeman (2002): https://doi.org/10.1080/13504620220145401). It's also not clear that the various survey findings reported are measuring the same thing and are therefore comparable. The survey section wording has been amended and less relevant ones removed.

The rationale for prioritising data / information from North America and English-speaking nations, and the limitations of this approach, warrants further discussion. This is particularly true for section 3.4 where the vast majority of examples are from North America. Is this a real effect, i.e. the influence of climate deniers on education really is found mainly in North America, or is this sampling bias? If the former, could / should this review focus specifically on North America as a location? Further, if the aim of this section is to present evidence for 'petro-pedagogy' are all of these examples really necessary, or is there value in presenting representative examples of differing approaches? What criteria are used to identify content as 'climate denial'? While this appears quite blatant / direct in some examples, in others it's much more subtle / indirect. I sought to list climate denial, direct and petro-pedagogy, wherever I could find it in schools. The definition of climate denial used is mentioned in the methodology. I was not deeply concerned with different approaches with the various organizations (though they are usually noted) as the goal is the same: no legislation to hinder the use of fossil fuels. In regards to the focus on English-speaking nations, more literature is available (see, for example, https://www.theguardian.com/environment/blog/2011/nov/11/climate-change-scienceofclimatechange#comment-13244224).

5) Do the authors give proper credit to related work and clearly indicate their own new/original contribution?

All sources are appropriately referenced. The contribution is more of a 'call to arms' than presentation of new knowledge.

6) Does the title clearly reflect the contents of the paper?

Not entirely. It should indicate, in some way, that this is not a complete review. Added to the abstract that the review was mainly restricted to English-language sources.

7) Does the abstract provide a concise and complete summary?

It would be helpful to indicate the limitations of the information considered in the review, e.g. geographical extent. See 6

8) Is the overall presentation well structured and clear?

At >12,000 words of body text the paper is too long. Almost half of this content is in Section 3.4. While the journal does not give guidance on manuscript length the aim of a review article should be to 'summarise the status of knowledge', and this is not a summary so consider where information could be streamlined, synthesized, tabulated, and / or moved to an appendix of supplementary information.  I felt that a compilation of the organizations was the focal point of the review to illustrate the scope.

9) Is the language fluent and precise?

I found the overall tone of the paper to be somewhat confrontational. I suspect this is deliberate and, to be fair, it can be quite effective at hooking the reader's attention. I do, however, question whether this style of writing is appropriate for an academic journal as it detracts from the narrative, and risks alienating readers who genuinely want to engage with the content.  It is deliberate. At this stage, there is no indication that hard-core climate deniers, who lead this movement, wish to honestly engage, and time is running out.

10) Are the number and quality of references appropriate?

The author references an impressive body of information and literature which could be separated into literature informing the main narrative and literature forming the 'results' from the review process. Despite the high number of references there are multiple places in the text where key information is unreferenced. Please share such references, if not mentioned below.

Further comments:

L15: 'Summarise status of knowledge' seems more appropriate than 'draw attention to'. done

L19: How is 'climate education' defined? Is this the same as 'climate change education'? Yes

L29: I disagree that this review is global in extent. It would be more appropriate to indicate from the outset that the information reviewed originates mainly from North America and English-speaking nations.

L45: Another reference for climate denial to consider is Jacques (2012): https://doi.org/10.1162/GLEP_a_00105 The sentence was removed.

L48-49: Please provide a reference for the tobacco industry example. done

L52-57: The phrase 'Activism opposing climate denial has arisen' needs further substantiating. Where dates are provided these are all very recent – there a particular timescale of interest? The paragraph now begins with - Opposition to climate denial generally has arisen, but is only briefly mentioned in this review. Popular (and my favourite) examples of climate activism include

L61: What is the timescale of 'recently'? This sentence was removed.

L63: Provide references for these previous studies. Paragraph replaced with -

Their involvement in schools became more of a concern after climate change was found to be caused by the burning of fossil fuels. A referee of this review explained that the excursion of climate denial into the classroom (in America) was: "basically backlash to the inclusion of climate change in K-12 education, which is probably less than 20 years old in the US, with a surge after the release of the NGSS [Next Generation Science Standards in 2013; see below] (RC1, 2024, 753-754)". For recent reports in America, for example, where the largest number of climate denial organizations in schools were found, see Atkin, 2020; Climate Town, 2023; Damico and Baildon, 2022; Noor and Westervelt, 2023; Reid and Branch, 2023; Strauss, 2017; Waldman, 2023 a-c; Worth, 2021a; and Zou, 2017.

Climate denial in the classroom is the focus of this review, which provides a summary of the climate-denial organizations that are the leading offenders in manipulating climate education in schools. An important general goal of the review is to create awareness of the growing threat in the classroom, so that teachers and parents can protect children in their schools from the anti-science influences of climate denial, and climate education researchers and instructors are aware of this menace. Is climate denial in your school, or your child's school, or in any local school?

L64: See earlier comment – to what extent can public knowledge be considered a reliable indicator of education? What is the evidence for this? Comment removed.

L70-71: Is there a reference for this statement about climate education? See above for new Method.

L72-77: see previous comment about methods. Expanded method section as shown above.

L79-90: I would expect the study questions to appear before the method, i.e. define questions, then state how the info required to address questions will be collected. Findings should not be stated at this point. I had placed the findings into the questions, but these have been removed, as shown above.

L83-85 / L91 / L98: What is recent? What is historical? Timescales need to be more precisely stated. Chose recent to be since 2021, which was added.

L87: "Most cases were found in America" - what was the sampling strategy used to locate examples? Described further in the methods above.

L98-99: "such polls are also indicative of the general state of climate education itself". This needs to be explained and referenced. Personally I disagree, but am open to being convinced. Sentence changed to: Recent surveys (since 2021) have revealed an alarming lack of understanding of the science of climate change by the public.

L99: How many polls were identified? How many have been considered for this review? I chose the polls that I was familiar with, without doing a detailed search, as results were generally consistent among these recent polls.

L109: The statement about climate education is confusing. When would this education take place? Once a politician is in office? This last paragraph was changed to -

The IPCC assessments, for example, demonstrate that the climate scientists know "very well" that climate change is occurring and the causes (2023, 4); the Pew survey shows that less than half of Americans are aware of the scientific consensus on climate change.

L110: Which Pew survey? L103 suggests that there are multiple surveys. The survey above as this section is part of the same paragraph.

L123: "The "alarmed" category matches best with the consensus on the science of climate change". This needs further explanation. On the suggestion of the other reviewer, I added the "Concerned" category. The consensus is mentioned in L109 above.

L134: Is there a reference to validate that these are "common climate denial talking points"? This sentence was removed.

L135: "Many Americans are clearly not familiar with…" Or choose not to believe? Sentence removed.

L139-143 (and other places): Why is US political orientation the only demographic variable discussed? It is a dramatic difference and is the most important factor to get legislation passed on climate change.

L155: Is scepticism the same as denial? The former is about doubt, the latter is more definitive. Sentence was removed.

L157-158: Unclear - 75% of countries, or average 75% of participants? Ditto L163 "77% agreed…" Upon the suggestion of the other reviewer, L153-186 were removed.

L173-178: Unclear how this information is relevant to the preceding info. See above.

L183: What questions were asked to ascertain public perceptions of the climate crisis? See above.

L188: I'm intrigued by the involvement of Meta in global surveys on climate change (just a comment, no response required).

L203: Specify recent climate change (as opposed to over geological time). Added - modern.

L205-209: See previous comment.  Added – modern.

L212: If climate education became a treaty obligation in 1992 then we might expect to see the influence of this in Millennials and later generations, but not earlier generations. So how is this captured in the survey data presented in 3.1 (or is it)? Sentence removed.

L237: Successful in what sense?  Education changed to - communications.

L290-1: Please provide references. Added - (Oreskes and Conway, 2010, Chapter 6; McCright and Dunlap, 2011; Dunlap and McCright, 2015; Lewandowsky, 2021; Kutney, 2024).

L308-314: This sounds like the value-action gap. See previous reference to Kollmus & Agyeman (2002), also Bushell et al. (2017): https://doi.org/10.1016/j.erss.2017.04.001. The latter reference was added.

L314-22:  What are the implications here for climate education? Studies that showed how to deal with climate deal generally.

L323-324: "Climate denial arises from fear of science messages about climate change, especially among conservatives. Political ideology plays a lesser role in climate legislation outside the United States" - but it still plays a role? This feels very dismissive of other locations, and the US-centric theme continues in the next paragraph. As a non-US resident I'm really not sure why I should care about this.  This certainly was not my intent, if anything, it was meant as a compliment to those outside the US who had not fallen so deeply for political polarization and for this propaganda movement. The paragraph just illustrates that this problem is worse in the US.

L330: Were the right-wing think tanks studied by McCright and Dunlap all from the US, or a range of locations? Their study focused on the US.

Section 3.4: please refer to the comprehensive comments provided by RC1 – I have nothing further to add to these. Below are my replies to RC1 -

359: What evidence is there for the comparative judgment ("more dire")? More dire in what aspects? In some ways, the situation is less dire, at least in the U.S., as shown by better treatment of climate change in state science standards and better preparation of science teachers between 2012 and the present. Sentence changed to - Over a decade later, the campaigns promoting climate denial in the classroom have escalated, as discussed in this section.

361-372: "softer" in what respect? It's clear enough that the most prevalent forms of climate change denial have been softening -- moving away from denying "it's real" and "it's us" and toward denying "it's bad" (to use Maibach's formulation) -- in general, and there's evidence that this is true of climate change denial campaigns targeting K-12 science education in the U.S. But the groundwork hasn't been laid in this article to discuss this transition here. This paragraph is also hard to follow in the absence of concrete examples. First sentence changed to - Climate denial in the classroom includes "petro-pedagogy". The term had been used to describe the energy-industrial complex funding energy and climate education programs for K-12 education, especially in STEM (science, technology, engineering, and mathematics) education (Eaton and Day, 2020, 462).

364: Likewise, "traditional conservative climate denial" hasn't been defined, so the contrast here will not be understood except by a reader already familiar with the situation. See above.

364: The groundwork hasn't been laid for the idea that the energy-industrial complex is providing K-12 science educational content (which moreover needs to be distinguished from the content provided by climate change denial organizations such as the Heartland Institute: see comment on 65-66). See above.

374: There are different relations being obscured by the word "sponsor" here: for example, actual fossil fuel companies provide the budget of the Oklahoma Energy Resources Board, but the Heartland Institute hasn't received directly traceable funds from such companies for a long time now, and many of its corporate sponsors made a point of cutting ties with them after the Unabomber billboard fiasco. It's misleading not to distinguish the different relations. Last paragraph changed to - Below is a description of climate-denial organizations and petro-pedagogy organizations promoting climate-denial in the classroom (also see figure 2 for editorial cartoons reflecting conservative climate denial views and liberal views of petro-pedagogy).

374: The parenthetical description doesn't capture the fact that the lefthand cartoon in figure 2 reflects conservative views of climate education while the righthand cartoon reflects liberal views of "petro-pedagogical" efforts. See above.

380-393: (1) What evidence is there that this program is global in actual reach? (Anybody can put material on the internet; that doesn't mean that it's global.) (2) This is under the rubric "Climate Denial in the Classroom" (the title of sec. 3.4); from the description this program is at worst soft denial. Does it make sense to begin or to spend a lot of time with the less pernicious examples? I placed groups globally if that is what they claimed. This group does fit the definition of petro-pedagogy.

394-409: The same comments apply as for 380-393. Also, is there any information on the content of Switch Classroom in particular? I agree that both groups appear to be less known, but they are trying. I wanted to flag as many groups as possible.

430-442: Is there any information on the impact on EU science classrooms? You made this good point above, on what is their impact; and that is a question that I found no answer, but they are a threat as they promote propaganda in schools.

444-472: Is there any information on current impacts? From what's said here, it sounds like these efforts wrapped up about ten years ago (unless the festival was later -- it's hard to tell; it may be a yearly event). Not recently, but I did add dates to the statements.

475-481: This section seems to be trying both to introduce the Canada material and to discuss the Saskatchewan episode; it would be clearer if the latter were moved into its own section (and if the dependence of the Canadian economy on extraction industries was discussed further). This paragraph was intended to be more an introduction to petro-pedagogy and was moved to section 3.4 as an introduction (with suitable modifications).

483-494: Interesting but not a lot of details about the actual content or uptake. Some websites did not provide many details, unless you signed up as a student or a parent.

495-502: The same comments apply as for 483-494. And if the worst aspect is that it promotes only personal action on climate, it's fairly soft denial. Soft yes, but a standard ploy of the fossil fuel industry to place the onus about the climate crisis on us and not them.

503-506: There's so little information here that it's hardly worth including. I wish to include them just to warn readers of this new group.

507-511: The same comment applies as for 503-506. Same as above.

531-534: This shouldn't be in the Ten Peaks section: either in the introductory Canada section or a concluding section on its own. (Or perhaps the blank line 529 is supposed to set it off? Fair if so.) Yes, an extra line was added to separate it, as a short summary.

536-541: (1) Yale has had a series of surveys with this question, with a bit of up and down but generally in the mid-to-high 70s. (2) Other surveys have addressed the issue with different questions, confirming the high level of support but offering further insights that may be worth discussing here, e.g.:

Kamenetz A. 2019, April 22. Most teachers don't teach climate change; 4 in 5 parents wish that they did. National Public Radio.

Pizmony-Levy O, Pallas, A. 2019. Americans endorse climate change education. Teachers College, Columbia University. https://www.tc.columbia.edu/media/centers-amp-labs/the-public-matters/AMERICANS-ENDORSE-CLIMATE-CHANGE-EDUCATION-final-version-posted-v09172019.pdf

Lange J. 2023, December 15. Poll: Americans overwhelmingly want climate change taught in schools. Heatmap. https://heatmap.news/climate/education-climate-trust-teachers-poll#

Your sharing of the references was most kind. I only added the last one, as it was most recent. I was not familiar with Heatmap; it is a nice site.

542-545: There have been multiple attempts to introduce the CCEA in both houses of Congress, most recently S. 4117 and H.R. 7946 (both introduced after the paper was submitted). Reference added, thanks for sharing.

546: Public education in the U.S. is ultimately controlled by the state, but the bulk of decisions on curriculum and instruction are made at the district level -- and there are about 13,500 local school districts -- or below (school, department, classroom). So there's even more decentralization than is revealed here. Role of district level was added.

546-548: (1) This is irrelevant to the decentralization point. (2) At face value, the campaign was deeply misconceived, not taking into consideration the facts that standards are revised on a multiyear schedule and that education policymakers are not likely to be responsive to single-shot petitions. (Of course, the ulterior motive may have been just to harvest addresses from people concerned about climate change education, and in this it may have succeeded.) Interesting, but I left it in to demonstrate that some groups were attempting to promote climate education at the state level.

550: 26 states were involved with (were "Lead State Partners" on) the development of the NGSS; not all adopted them (as correctly implied below). Added

557: the 24 (actually 25 now) states said to be "using [the NGSS] as guides" have actually based their standards on the same National Research Council Framework on which the NGSS are based -- not much difference in practice, but it makes a difference in some contexts. Comment on Pennsylvania added.

557-558: since 2020, PA moved from the non-Framework category to the Framework category, so only five states are neither NGSS nor Framework: TX, FL, OH, VA, and NC. Reference added, thanks.

560: re "not guided by the NGSS," see comment on 557 The change of Pennsylvania was noted.

575-579: More details about what the study sought to understand, and more specific reportage on its results, are needed. The study looked for state-level (i.e. state board of education or state department of education) policies regarding climate change in four contexts -- "1) institutional governance, 2) teaching and learning, 3) facilities and operations, and 4) community partnerships" -- of which only 2 is really relevant here. The new paragraph reads - A study of climate education reviewed: "802 publicly available education policies across the United States" and "used a whole institution approach for data collection and analysis and considered four institutional domains of potential climate change activity: 1) institutional governance, 2) teaching and learning, 3) facilities and operations, and 4) community partnerships (MECCE and NAAEE, 2022, 4-5)." Among their findings were that all states had policies mentioning climate change, but 33 states had very low focus, and 14 states had low focus on climate change content (9-11, 24, 40). States that followed the NGSS were more likely to include climate change content (9, 24-26). When energy was taught, there was little mention of climate change (9, 31-37). The report again highlighted the issue of climate denial: "For decades, political and social will to act on climate change was quickly swept away in a current of denial, avoidance, and political posturing (3; see also 7, 28, 44)".

592-607: It's surprising not to see any mention of the CO2 Coalition's recent attempt to disseminate its materials at a NSTA conference. See e.g. https://www.washingtonpost.com/climate-environment/2023/04/11/co2-coalition-climate-denial/ Thanks this reference. The following has been added - They describe themselves as: "comprised of more than 100 of the top experts in the world who are skeptical of a theoretical link between increasing CO2 and a pending climate crisis while embracing the positive aspects of modest warming and increasing CO2 (CO2 Coalition, 2023, 1). Among their activities is the development of programs on science education:

In early 2021, a group of CO2 Coalition members decided to act on their concerns about the state of science education in America. They recognized that the teaching of science had strayed from the 400-plus-year-old scientific method and was less inclined to encourage inquisitiveness in students and more prone to require conformity to the opinions of teachers. At present, much of the instruction on climate change resembles an indoctrination into a political agenda rather than the provision of necessary tools for critical thinking (2023, 1).

On March 23, 2023, they issued a booklet attacking the position of the National Science Teaching Association on climate change (CO2 Coalition, 2023), using standard climate-denial talking points. The final conclusions of the booklet were:

As a result, students are undergoing an indoctrination into a dangerous political agenda that ignores the enormous benefits of CO2 – a gas critical to life – and promotes an impossible objective of supporting modern economies without carbon-based energy sources.

We respectfully urge the National Science Teaching Association to seriously consider a rejection of their previous endorsement of scientific censorship and return science education to the foundations of reason, open scientific debate and tolerance for alternative thinking (CO2 Coalition, 2023, 16).

The booklet was released as the National Science Teaching Association was holding a convention, where the CO2 Coalition had a booth and distributed the booklet and a comic book "Simon the Solar-Powered Cat," depicting carbon dioxide as being good for the planet. An article in the Washington Post about the episode warned that the CO2 Coalition literature could cause teachers to spread propaganda about the science of climate change to their students (Joselow, 2023). The members of the CO2 Coalition were kicked out on the second day of the convention.

612: "yet" is gratuitous Deleted

618-651: Is there any evidence of uptake by teachers? This has not been investigated to my knowledge.

652: It's hard to know how to assess this claim. They certainly have their differences (e.g., CO2 Coalition is single-issue and the others aren't; EverBright Media is for-profit and the others aren't; etc.). Agreed but the propaganda they all share is typical climate denial talking points.

652: Why no mention in this context of PragerU Kids, which isn't an outlier compared to these three and whose climate change denial videos and comics aimed at kids grabbed headlines in 2023 in a number of states, especially FL? See e.g. https://www.politico.com/news/2023/08/09/in-desantis-florida-schools-get-ok-for-climate-denial-videos-ee-00109466 PragerU was added

653-664: It would bear mention that ONEI is the OH equivalent of the OERB. Added to section on the OERB.

665-681: (1) The bill was amended to refer to climate policy rather than climate change, so it was transformed from hard to soft denial. (2) As acknowledged, this bill isn't about K-12 education, so why include it? (3) The bill hasn't passed or been enacted, so why include it, especially when a number of bills that actually sought to undermine K-12 climate change education in various states aren't discussed? Removed

682-695: It would bear mention that OERB was the first of its kind, with ONEI and others inspired by it. Why no mention of e.g. the Illinois Petroleum Resources Board? See e.g. https://www.levernews.com/a-fossil-fuel-miseducation/ Added -

3.4.3.2.4 Illinois Petroleum Resources Board

The goal of the Illinois Petroleum Resources Board is to "improve the image and credibility of the Illinois oil and gas industry", and this is accomplished through seven objectives, of which the first is "Education: Create an understanding of the Illinois oil and gas industry and good safety practices through programs with schools, organizations and the public at large (Illinois Petroleum Resources Board, undated a)". They offer a series of professional development programs for middle and high school teachers (Illinois Petroleum Resources Board, undated b). No connections between fossil fuels and climate change were found on their website. Blogs on their website generally defended petroleum, including one titled "Benefits of Fossil Fuels to Humanity Have Far Outweighed Negatives":

But it is dangerously misleading to focus exclusively on those [environmental] impacts and completely ignore their massive benefits. And using this deeply flawed framing as the basis of campaigns to rapidly eliminate the source of 83 percent of the world's energy and virtually all our modern products is even more dangerous considering the favored "alternatives" are completely inadequate to replace fossil fuels.

It can't be emphasized enough that renewable energy – specifically wind and solar – can only generate electricity, and do so only when the sun is shining and the wind is blowing (Whitehead, 2022).

An article criticizing the Illinois Petroleum Resources Board was called "A Fossil Fuel Miseducation", which stated: "the IPRB doesn't appear to deny climate change — they mostly seem

to avoid mentioning it at all. Instead, the group focuses on economic arguments about the oil and gas industry, which they claim will be a good source of jobs for decades to come, despite mounting evidence to the contrary (Gopal, 2024)".

698: Not "an education vendor"; rather, the DOE gave its imprimatur to the use of PragerU Kids videos in middle school social studies classes. Importantly, (1) it would be hard to justify the use of its climate change denying videos on those classes, given the lack of climate change content in the corresponding state standards, and (2) the DOE's imprimatur is basically irrelevant, since districts make decisions about instructional material, and some of the bigger districts have already said that they will not allow PragerU Kids materials to be used. Changed "an education vendor" to - allowed in Florida schools.

701: These states took different actions on different PragerU Kids products; it's misleading to lump them together as all approving the use of climate change denial material.

* In MT, the state superintendent of public instruction, Elsie Arntzen, a Republican, signed a textbook license agreement with PragerU. This doesn't have much actual significance, because the only requirement for obtaining such a license is posting a surety bond -- in PragerU's case, for $5000. Having the license doesn't mean that instructional materials will be considered, let alone approved, and the superintendent doesn't make decisions on instructional materials anyhow. But Arntzen expressed her approval of the materials, which might conceivably have some effect on districts' decisions.

* In NH, high school students in New Hampshire now have the opportunity to satisfy their financial literacy graduation requirement with PragerU Kids's Cash Course module online. The commissioner of the state department of education, Frank Edelblut, a Republican, even appeared in a promotion for it. Whatever you think about this, it's not likely to have any effect on climate change education.

* In OK, the state Department of Education under state superintendent of public instruction Ryan Walters, a Republican, announced a "partnership" with PragerU, which seems to take the form of the department endorsing its videos for use in social studies classrooms. While there are opportunities to discuss climate change in social studies classes, Oklahoma's social studies standards don't provide a lot of opportunities for it, so the effect on climate change education will probably be limited. The Oklahoma Education Association reacted to Walters's announcement by reminding districts that they don't have to use the material and parents that they could opt their children out of exposure to them.

\* In TX, a PragerU promotion that featured praise from Julie Pickren, a Republican member of the state board of education, claimed that PragerU is an approved education vendor in the state. That was not, and still is not, actually true. Texas has a lot of problems with climate change education, thanks in part to Pickren, but official approval of PragerU for Kids materials is not among them. I changed the wording to - other states expressed interest in the PragerU programs.

703-709: It's surprising not to see mentioned the not-very-hidden analogy Jews in the Warsaw Uprising:Nazis :: climate change deniers:climate change accepters. Not that I disagree, but I prefer to stay away from this comment, as it invites new criticisms from climate deniers.

726: It's actually a lesson plan for grades 3-5, and it has API branding. While it can be read as soft denial, it's intended to be a career lesson, not an environmental science lesson. Changed to - lesson plan for grades 3 to 5.

729: the heading should be "Texas State Board of Education" -- the TEA is basically the TX department of education; it's administrative and doesn't set policy and can't be blamed for the shenanigans discussed in this section Changed as recommended

729-739: (1) This is state action, so it doesn't seem like it belongs in this section. (2) There have been similar episodes elsewhere, with executive and legislative actions aimed at inhibiting climate change education and apparently motivated by climate change denial; why aren't they discussed as well? The Texas case has attracted much attention and has implications for publishers generally.

737-739: (1) This was not "unrelated"; the changes to the board operating rules were made in order to facilitate the later attacks on the textbooks. (2) The textbooks in question were not banned; they were not approved. Districts are still free to use them if they wish; it's just harder and more expensive for them to do so. (3) The textbooks were not climate textbooks but grade 8 science textbooks. Last lines changed to - Textbook censorship in Texas, and other states, are increasing. One local school district in Houston voted to censor chapters on climate change; a local parent feared: "It's really kind of alarming what this could mean for ideological influence and control over what is taught in schools (Salam, 2024; see also 2023)".

740-746: This isn't part of section 3.4.3.2.8, and should be set off from it somehow. Shortened and made part of the section 1.

L750: Is 'misunderstanding' the right term to describe the Consensus Gap? I think for some people this is a conscious choice. Agreed, the sentence was removed.

L757-758: "Climate education, despite a serious and genuine effort, has failed to teach the world about the causes and risks of the climate crisis". To what extent can this be generalised to 'the world', given that the vast majority of the evidence presented relates to North America? Sentence removed.

L759-65: I'd really like to see these recommendations for IPCC publications focused on climate denial, and aimed at alternative audiences, followed through.

L766-68: I'd be interested to know social scientists' view on this! I would be as well. As stated, this is my opinion looking at this through the eyes of a climate activist challenging climate denial for the past decade.

L775-7: "Climate education has been relatively successful with liberals but has had no impact on conservatives in some countries for more than a decade". Please provide evidence / references to support this. This paragraph was removed based on the comments of RC1.